# W-PCA Based Gradient-Free Proxy for Efficient Search of Lightweight Language Models

**Shang Wang**
ShanghaiTech University
`wangshang2024@shanghaitech.edu.cn`

## Abstract

The demand for efficient natural language processing (NLP) systems has led to the development of lightweight language models. Previous work in this area has primarily focused on manual design or training-based neural architecture search (NAS) methods. Recently, zero-shot NAS methods have been proposed for evaluating language models without the need for training. However, prevailing approaches to zero-shot NAS often face challenges such as biased evaluation metrics and computational inefficiencies. In this paper, we introduce weight-weighted PCA (W-PCA), a novel zero-shot NAS method specifically tailored for lightweight language models. Our approach utilizes two evaluation proxies: the parameter count and the number of principal components with cumulative contribution exceeding $\eta$ in the feed-forward neural (FFN) layer. Additionally, by eliminating the need for gradient computations, we optimize the evaluation time, thus enhancing the efficiency of designing and evaluating lightweight language models. We conduct a comparative analysis on the GLUE and SQuAD datasets to evaluate our approach. The results demonstrate that our method significantly reduces training time compared to one-shot NAS methods and achieves higher scores in the testing phase compared to previous state-of-the-art training-based methods. Furthermore, we perform ranking evaluations on a dataset sampled from the FlexiBERT search space. Our approach exhibits superior ranking correlation and further reduces solving time compared to other zero-shot NAS methods that require gradient computation.

## 1 Introduction

Large language models (LLMs) have shown exceptional performance across various domains (OpenAI, 2023). However, their size and computational demands pose challenges in resource-constrained environments like mobile devices and edge computing. Therefore, there is a growing need to explore lightweight language models that can operate efficiently on these platforms. One approach to address this challenge is through knowledge distillation (KD) (Liu et al., 2023; Li et al., 2023c; Li & Jin, 2022; Li et al., 2023b; Li, 2022; Li et al., 2024a), where a larger language model acts as a teacher to train a smaller, more lightweight language model (Turc et al., 2019; Sanh et al., 2020; Jiao et al., 2020; Sun et al., 2020; Wang et al., 2020). However, the student models trained for these tasks were manually designed. To effectively search for student models, the use of neural architecture search (NAS) has become essential.

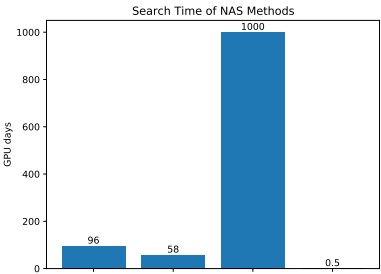

Figure 1: Comparison of the running time between W-PCA and other training-based NAS methods for lightweight language models. Our method achieves a substantial reduction in search time for the optimal network structure by two to three orders of magnitude, as we do not need to train the supernet.

NAS is a technique that automates the process of designing neural networks, enabling the exploration of a wide range of architectures to identify the most optimal ones for a given task. Vanilla NAS approaches primarily used reinforcement learning (Zoph & Le, 2016) or genetic algorithms (Real et al., 2019) to train neural networks from scratch,

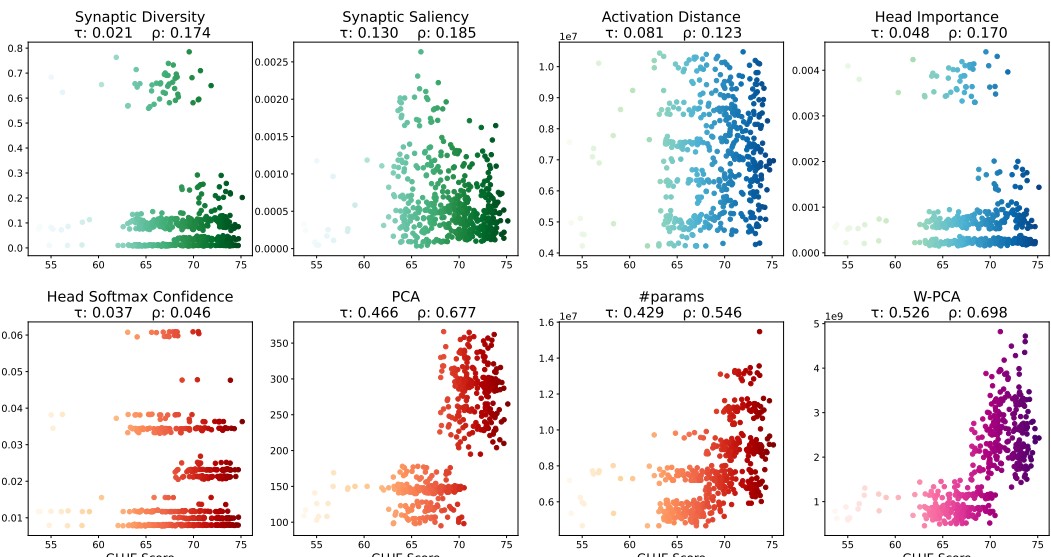

Figure 2: Plots depicting the evaluation of zero-shot proxy metrics on 500 randomly sampled architectures from the FlexiBERT search space. As in literature (Serianni & Kalita, 2023), we use the GLUE score of each neural network as the ground truth and evaluate the performance of each zero-shot proxy metric based on its ranking correlation with the ground truth. The specific calculations of PCA is described in Section 3.2, and the respective zero-shot proxies used for the comparison are summarized in Section 2.3. Our metric W-PCA is calculated as the product of the number of parameters (#params) and the principal component analysis (PCA).

but these methods were computationally expensive. Subsequently, one-shot NAS methods, such as gradient-based (Liu et al., 2018) and single path one-shot (SPOS) methods (Guo et al., 2020), were proposed. These methods are more efficient as they leverage pre-trained models or parameter sharing, requiring the establishment of a supernet in advance from which to sample the optimal subnetworks. Importantly, many lightweight model search tasks in natural language understanding (NLU) are accomplished using one-shot NAS (Xu et al., 2021; Dong et al., 2021; Gao et al., 2022). While one-shot NAS reduces training costs compared to training from scratch, it still requires various training strategies to effectively train the supernet. However, to further enhance search efficiency, it is necessary to introduce zero-shot NAS (Mellor et al., 2021a). Zero-shot, also known as training-free NAS, is a promising approach that eliminates the need for training neural networks and directly evaluates their performance using proxy metrics. This significantly reduces the training time and computational resources required for NAS.

Existing zero-shot NAS methods (Abdelfattah et al., 2020; Wei et al., 2024; Dong et al., 2023a;b) have primarily focused on ranking correlations on NAS benchmark datasets (Klyuchnikov et al., 2022), with limited consideration for specific deep learning tasks. This limitation hinders their applicability and effectiveness in practical scenarios. Additionally, these methods often solely consider a single feature of the models, leading to biased evaluations and potentially overlooking important characteristics.

In our research, we aim to address these limitations and improve the applicability of zero-shot NAS. In Section 5, we conducted ranking correlation experiments. As illustrated in Figure 2, our attempts to incorporate previous zero-shot proxies into language model evaluation yielded unsatisfactory results. However, we observed a strong correlation in ranking between principal component analysis (PCA) and the number of parameters (#params), with their product demonstrating even better performance.

Motivated by these findings, we propose a novel approach called Weight-Weighted PCA (W-PCA), which takes into account both the parameter count and the number of principal components with cumulative contribution exceeding a given $\eta$ threshold in the model. By integrating these two factors, our aim is to achieve a more accurate evaluation of language models in the context of zero-shot NAS. Furthermore, we have designed a search space specifically for NLU tasks and applied our designed zero-shot proxies, as well as the previous zero-shot proxies used in Transformer, to this search space. To the best of our knowledge, this is the first work that applies zero-shot NAS to NLU tasks.

## 2 RELATED WORK

### 2.1 LIGHTWEIGHT BERT MODELS

Turc et al. observed that distillation and pre-training + fine-tuning have mutually reinforcing effects. DistilBERT (Sanh et al., 2020) utilizes a triple loss function for training the lightweight model. TinyBERT (Jiao et al., 2020) applies distillation in both the pre-training and task-specific learning phases. MobileBERT (Sun et al., 2020) proposes a bottleneck structure to reduce the parameter count. MiniLM (Wang et al., 2020) introduces a compression method called deep self-attentive distillation. In this study, we incorporate both the standard BERT-base (Devlin et al., 2019) and MobileBERT models, along with their weight-sharing variations, where each layer is integrated into the supernet.

### 2.2 ONE-SHOT NAS FOR EFFICIENT MODELS

Numerous methods have been proposed for performing neural architecture search (NAS) (Hu et al., 2021; Dong et al., 2022; Sun et al., 2024; Li et al., 2024d;c) to develop efficient models. NAS-BERT (Xu et al., 2021) trains a large supernet on a carefully designed search space that includes diverse architectures, generating multiple compressed models with adaptable sizes and latency. EfficientBERT (Dong et al., 2021) proposes a three-stage coarse-to-fine search scheme to optimize the combination of the multilayer perceptron (MLP) in the feed-forward network (FFN), ultimately reducing the parameter count of the FFN. AutoBERT-Zero (Gao et al., 2022) devises a search space that includes unary and binary math operators for constructing attention structures and backbones for general pre-trained language models (PLMs) from scratch. To the best of our knowledge, it is the most recent NAS method that incorporates lightweight BERT models in the experiment.

### 2.3 ZERO-SHOT NAS

Zero-shot NAS has been applied to transformer-based architectures in several ways. We provide a summary of these applications below.

**Synaptic Saliency** (Tanaka et al., 2020) aims to prevent layer collapse during network pruning, as this collapse can significantly reduce the accuracy of the network. The formulation for this approach is expressed as follows:

$$S(\theta) = \frac{\partial \mathcal{L}}{\partial \theta} \odot \theta$$

where $\mathcal{L}$ represents the loss function, $\theta$ denotes the network's parameters, and $\odot$ is the Hadamard product. Abdelfattah et al. generalize synaptic saliency as a zero-shot metric for NAS by summing over all $n$ parameters in the network: $S = \sum_{i=1}^{n} S(\theta_i)$

**Synaptic Diversity** builds upon previous research on rank collapse in transformers. In this phenomenon, the output of a multihead attention block tends to converge to rank 1 for a given set of inputs, which significantly impairs the performance of the transformer. Zhou et al. propose a method that utilizes the nuclear norm of an attention head's weight matrix $W_m$ as an approximation of its rank. This approach leads to the computation of the synaptic diversity score as follows:

$$S_D = \sum_m \|\frac{\partial \mathcal{L}}{\partial W_m}\|_{nuc} \odot \|W_m\|_{nuc}$$

**Activation Distance** is a proxy metric introduced by Mellor et al. to assess the ReLU activations of a network. By computing the Hamming distance between the activations within the initialized network for each input in a minibatch, this metric determines the similarity of the activation maps. The authors observe that when the activation maps for a given set of inputs exhibit higher similarity, the network faces greater difficulty in disentangling the input representations during the training process.

**Jacobian Covariance** evaluates the Jacobian $J = \left( \frac{\partial L}{\partial \mathbf{x_1}}, \ldots, \frac{\partial L}{\partial \mathbf{x_N}} \right)$ of the network's loss function with respect to the minibatch inputs. Further details of this metric can be found in the original paper (Mellor et al., 2021b).

**Jacobian Cosine** (Celotti et al., 2020) is proposed as an improvement to the Jacobian Covariance metric, aiming to enhance computation speed and effectiveness. This improvement involves utilizing

cosine similarity instead of a covariance matrix to measure similarity. The metric is computed as follows:

$$S = 1 - \frac{1}{N^2 - N} \sum_{i=1}^{N} |J_n J_n^T - I|^{\frac{1}{20}}$$

Here, $J_n$ represents the normalized Jacobian, and $I$ is the identity matrix. The metric is computed using a minibatch of $N$ inputs. In their large noise and more noised scores, the authors introduce various noise levels to the input minibatch, hypothesizing that architectures exhibiting high accuracy will demonstrate robustness against noise.

**Attention Confidence, Importance, and Softmax Confidence** "Confident" attention heads exhibit high attention towards a single token, indicating their potential importance to the transformer's task. Researchers have proposed different approaches to calculating confidence, including examining the softmax layer of the attention head and analyzing the sensitivity of the attention head to weight masking by computing the product between the attention head's output and the gradient of its weights. Serianni & Kalita summarize the findings from (Voita et al., 2019; Behnke & Heafield, 2020; Michel et al., 2019) regarding the following metrics:

Confidence: $A_h(\mathbf{X}) = \frac{1}{N} \sum_{n=1}^{N} |\max(Att_h(\mathbf{x}_n))|$

Softmax Confidence: $A_h(\mathbf{X}) = \frac{1}{N} \sum_{n=1}^{N} |\max(\sigma_h(\mathbf{x}_n))|$

Importance: $A_h(\mathbf{X}) = |Att_h(\mathbf{X}) \frac{\partial \mathcal{L}(\mathbf{X})}{\partial Att_h(\mathbf{X})}|$

where $X = \{x_n\}_{n=1}^N$ represents a minibatch of $N$ inputs, $\mathcal{L}$ denotes the loss function of the model, and $Att_h$ and $\sigma_h$ denote an attention head and its softmax, respectively. To obtain an overall metric for the entire network, Serianni & Kalita extend these scores by averaging them across all $H$ attention heads: $\mathcal{A}(\mathbf{X}) = \sum_{h=1}^{H} \frac{1}{H} Att_h(\mathbf{X})$

## 3 OUR GRADIENT-FREE WEIGHT-WEIGHTED PCA PROXY

### 3.1 MOTIVATION FOR USING PCA

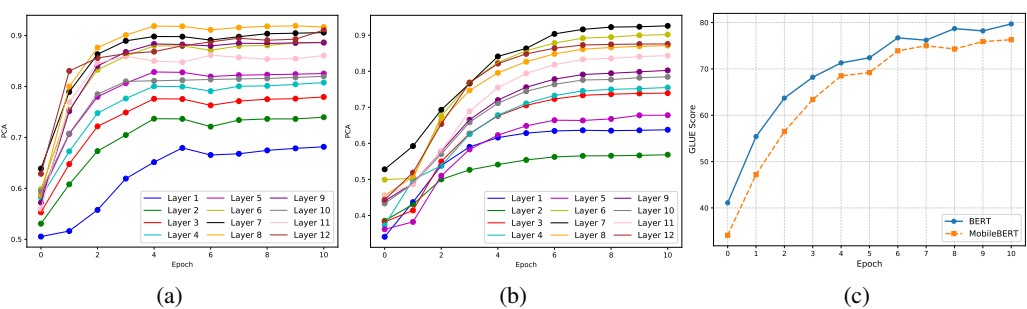

(a)  (b)  (c)

Figure 3: (a) and (b) show the PCA score curves for BERT (Devlin et al., 2019) and MobileBERT (Sun et al., 2020), respectively, at different epochs during training ($\eta$=0.99). (c) presents the progression of GLUE scores for BERT and MobileBERT over training epochs.

Inspired by the work in Pan et al. (2023), which leverages PCA to optimize the training of Vision Transformers, we analyzed the trends in PCA variation during the training of BERT and MobileBERT. From Figure 3, we can draw the following conclusions:

1. *Tracking performance via PCA.* As shown in Figure 3, the overall GLUE score and the PCA values for each layer progressively increase with the number of training epochs. This indicates that PCA values effectively reflect the performance of the neural network. Specifically, the steady rise in PCA values suggests that the network's internal representations become more structured and discriminative.

2. *Diminishing returns after peak performance.* Our observations also revealed that after reaching peak values, the PCA curves flatten out, indicating that further training yields diminishing returns.

This aligns with the conclusions in Table 13, where we discussed that prolonged training results in minimal performance gains.

3. *Early PCA values as predictors.* Notably, layers with higher PCA values at epoch 0 (i.e., before training begins) tend to maintain higher PCA values throughout training. This led us to hypothesize that the initial PCA values could serve as effective indicators for comparing neural network architectures during the NAS phase.

To validate these hypotheses, we conducted experiments as shown in Figure 2, which confirmed that early PCA values correlate well with the final performance ranking of the networks. This insight motivated us to further pursue accuracy comparison experiments, yielding promising results.

## 3.2 VANILLA PCA PROXY

For a given $\eta$, the distribution of PCA principal component values reflects the proportion of useful information in the matrix. We attempt to use this distribution as a metric for evaluating neural network performance. Our metric is computed as follows:

$$S_f(\mathbf{X}) = \text{PCA\_dim}(\mathbf{X}, \eta) \tag{1}$$

Here, $\mathbf{X} \in \mathbb{R}^{B \times N \times D}$ represents a minibatch of inputs, where $B$ represents the batch size, $N$ is the token length, and $D$ represents the embedding dimension. $\eta$ represents the cumulative contribution rate of principal components. We calculate the PCA values for the hidden states after the initial linear transformation in the FFN layer. Specifically, if we express the FFN layer as:

$$\text{FFN}(\mathbf{X}) = \sigma(\mathbf{X}\mathbf{W_1} + b_1)\mathbf{W_2} + b_2 \tag{2}$$

Then, we compute the PCA value of the $\mathbf{X}\mathbf{W_1} + b_1$ part, denoted as $\mathbf{H} \in \mathbb{R}^{B \times N \times D'}$ in the following text, where $D'$ represents the hidden dimension. To compute the PCA values, we first reshape $H$ into a two-dimensional matrix $\mathbf{H}' \in \mathbb{R}^{(BN) \times D'}$. Then, we subtract the mean of each feature (column) from the data to center it:

$$\mathbf{H}_{centered} = \mathbf{H}' - \mathbf{H}'.mean(0) \tag{3}$$

We then calculate the covariance matrix of the centered data as:

$$\mathbf{C} = \frac{1}{(BN) - 1}\mathbf{H}_{centered}^T\mathbf{H}_{centered} \tag{4}$$

Next, we perform an eigenvalue decomposition on the covariance matrix $\mathbf{C} \in \mathbb{R}^{D' \times D'}$ to obtain eigenvalues ($\mathbf{\Lambda}$) and eigenvectors ($\mathbf{V}$):

$$\mathbf{C} = \mathbf{V}\mathbf{\Lambda}\mathbf{V}^T \tag{5}$$

Here, $\mathbf{\Lambda}$ is a diagonal matrix of eigenvalues and $\mathbf{V}$ is the matrix of eigenvectors. After sorting each eigenvalue $\lambda_i$ in descending order, we determine the minimum number of eigenvectors $k$ required to explain at least $\eta$ variance:

$$k = min\{k' | \frac{\sum_{i=1}^{k'} \lambda_i}{\sum_{i=1}^{D'} \lambda_i} \geq \eta\} \tag{6}$$

This value of $k$ represents the required $\text{PCA\_dim}(\mathbf{X}, \eta)$. By analyzing PCA_dim (the dimensions with PCA values exceeding a threshold $\eta$), we can identify the dimensions that contain a higher amount of valuable information.

The metric for an $m$-layer neural network model is obtained by summing $S_f(\mathbf{X})$ over all layers, resulting in:

$$S(\mathbf{X}) = \sum_{f=1}^{m} S_f(\mathbf{X}) \tag{7}$$

Here, the metric $S_f(\mathbf{X})$ represents the PCA-based value for a specific layer $f$.

Finally, to compute the overall metric $S(\mathbf{X})$ for an $m$-layer neural network model, we sum the layer-specific metrics $S_f(\mathbf{X})$ over all layers. By utilizing this methodology, we can effectively assess the performance of candidate architectures based on their PCA values and identify the dimensions that contribute significantly to the valuable information in the hidden states.

### 3.3 WEIGHT-WEIGHTED PCA PROXY

It is extremely challenging to discover or design a proxy that outperforms weight parameters (#Params) in terms of stability and performance (Li et al., 2023a). After identifying PCA as an excellent proxy, multiplying it with #Params is a worthwhile attempt, as the additional computation time for a proxy is negligible compared to training neural networks. Thus, we propose a new metric called W-PCA, which quantifies the amount of valuable information captured by each dimension relative to the number of parameters in the architecture.

The W-PCA metric is computed as the product of the number of weight parameters ($w$) and the PCA value for each dimension. Mathematically, it can be expressed as:

$$\text{W-PCA}(\mathbf{X}) = w \times S(\mathbf{X}) \tag{8}$$

Advantages of our method include:

1. **Strong Correlation:** The W-PCA metric captures the relationship between the number of parameters and the valuable information in each dimension. This relevance is crucial in evaluating the efficiency and effectiveness of candidate architectures. By considering the PCA values, we can identify dimensions that contribute the most to the architecture's performance, allowing for informed decision-making during architecture search.

2. **Gradient-Free:** Unlike many traditional optimization methods that rely on gradients, our methodology is gradient-free. This eliminates the need for extensive backpropagation and derivative calculations, making the evaluation process more efficient and less computationally expensive.

3. **One forward propagation only:** Our methodology requires only forward propagation during the evaluation of candidate architectures. This simplifies the implementation and reduces the computational overhead, as it avoids the need for complex and resource-intensive operations such as backpropagation.

By leveraging the advantages of strong relevance, gradient-freeness, and the use of only forward propagation, our methodology based on the W-PCA metric provides an efficient and effective approach for training-free architecture search. It enables researchers and practitioners to evaluate candidate architectures based on their valuable information content relative to the number of parameters, facilitating the exploration of architecture design space and aiding in the development of more efficient and effective models.

## 4 SEARCH SPACE FOR NLU TASKS

To enhance the evaluation of W-PCA's performance on NLU tasks, we have meticulously crafted a search space. Drawing inspiration from SPOS (Guo et al., 2020), our search targets a model comprised of multiple layers, with each layer capable of being a lightweight BERT model. The hidden dimension of the FFN layer within each BERT block is determined through random selection. This deliberate randomness aids in exploring a diverse range of architectures during the search process.

## 5 RANKING EVALUATION

### 5.1 DATASETS

To assess the accuracy of the proposed proxy indicators for neural network evaluation, we employed a benchmark consisting of a well-trained BERT structure suggested by Serianni & Kalita as the

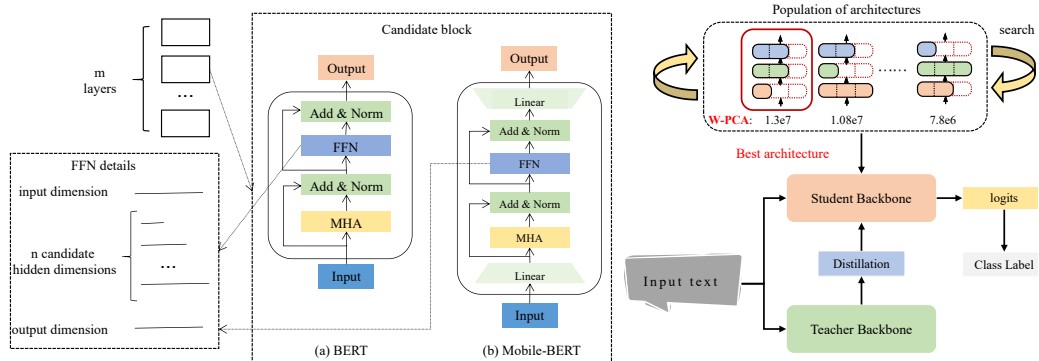

Figure 4: Overview of the W-PCA framework for NLU tasks. The search space consists of $m$ layers, each with 2 candidate blocks and $n$ candidate dimensions, resulting in a total of $(2 \times n)^m$ combinations. A genetic algorithm (detailed parameterization provided in Section 6.2.1) is employed to identify the optimal structure with the highest W-PCA value. This structure is subsequently refined through additional training using knowledge distillation (KD). In the figure, FFN and MHA represent the feed-forward network and multi-head attention, respectively.

testing dataset. Specifically, this benchmark selected 500 structures from the FlexiBERT (Tuli et al., 2023) search space (as presented in Table 7) and utilized ELECTRA (Clark et al., 2020), rather than the MLM method, for training to efficiently pretrain a compact BERT model. The training dataset comprised 8,013,769 documents sourced from the OpenWebText (Gokaslan et al., 2019) corpus, amounting to a total of 38GB. For detailed training information, please refer to Appendix B. After training, the scores obtained by fine-tuning on the GLUE dataset will serve as the reference for evaluating the correlation among different zero-shot proxies.

## 5.2 RESULTS AND ANALYSIS

Table 1: Comparison of different zero-shot proxies on the FlexiBERT benchmark. "Time" represents the computation time for the metric calculated 1,000 times. Both $\tau$ and $\rho$ are computed to measure the ranking correlation between each zero-shot proxy and the ground truth of the neural networks, represented by their BLEU scores

| Proxy | Time | $\nabla$-free | $\tau$ | $\rho$ |
|---|---|---|---|---|
| Synaptic Diversity (Zhou et al., 2022) | 110 s | ✗ | 0.021 | 0.174 |
| Synaptic Saliency (Abdelfattah et al., 2020) | 121 s | ✗ | 0.130 | 0.185 |
| Activation Distance (Mellor et al., 2021a) | 68 s | ✓ | 0.081 | 0.123 |
| Jacobian Cosine (Celotti et al., 2020) | 103 s | ✗ | 0.116 | 0.149 |
| Head Importance (Serianni & Kalita, 2023) | 112 s | ✗ | 0.048 | 0.170 |
| Head Confidence (Serianni & Kalita, 2023) | 81 s | ✓ | 0.306 | 0.364 |
| Vanilla PCA | 61 s | ✓ | 0.466 | 0.677 |
| W-PCA | 74 s | ✓ | **0.526** | **0.698** |

The evaluation results include the Kendall rank correlation coefficient (Kendall $\tau$) and the Spearman rank correlation coefficient (Spearman $\rho$). Table 1 demonstrates that Vanilla PCA has already surpassed the previous zero-shot proxy in terms of ranking correlation, and W-PCA performs even better than Vanilla PCA. Furthermore, W-PCA achieves higher computational efficiency than proxies due to the absence of gradient computation. In Appendix C, we compare the ranking stability of W-PCA with other zero-shot metrics using different initialization weights and batch inputs.

## 6 ACCURACY COMPARISION

### 6.1 DATASETS

To enable an accurate comparison to other lightweight BERT models, we evaluate the performance of W-PCA using the GLUE (Wang et al., 2018) and SQuAD (Rajpurkar et al., 2016) datasets and their corresponding task-specific evaluations.

Table 2: Performance comparison of the test set on the GLUE benchmark. The performance of all zero-shot proxies is evaluated on the search space depicted in Figure 4. Latency measurements of the models are conducted using the NVIDIA A100 GPU.

| Model | Type | #Params | Latency | QNLI | MRPC | SST-2 | CoLA | STS-B | MNLI-m/mm | RTE | QQP | AVG |
|---|---|---|---|---|---|---|---|---|---|---|---|---|
| BERT-base (Devlin et al., 2019) | manual | 108.9M | 274ms | 90.5 | 88.9 | 93.5 | 52.1 | 85.8 | 84.6/83.4 | 66.4 | 71.2 | 79.6 |
| BERT-base (ours) | manual | 108.9M | 274ms | 91.4 | 88.7 | 93.0 | 49.0 | 87.5 | 84.9/83.9 | 76.6 | 71.3 | 80.7 |
| BERT-tiny (Turc et al., 2019) | manual | 14.5M | 44ms | 84.8 | 83.2 | 87.6 | 19.5 | 77.1 | 75.4/74.9 | 62.6 | 66.5 | 70.2 |
| BERT-small (Turc et al., 2019) | manual | 28.8M | 79ms | 86.4 | 83.4 | 89.7 | 27.8 | 77.0 | 77.6/77.0 | 61.8 | 68.1 | 72.1 |
| DistilBERT-6 (Sanh et al., 2020) | manual | 67.0M | 151ms | 88.9 | 86.9 | **92.5** | **49.0** | 81.3 | 82.6/81.3 | 58.4 | 70.1 | 76.8 |
| TinyBERT-4 (Jiao et al., 2020) | manual | 14.5M | 45ms | 87.7 | 88.5 | 91.2 | 27.2 | 83.0 | 81.8/80.7 | 64.9 | 69.6 | 75.0 |
| MobileBERT-tiny (Sun et al., 2020) | manual | 15.1M | 62ms | 89.5 | 87.9 | 91.7 | 46.7 | 80.1 | 81.5/81.6 | 65.1 | 68.9 | 77.0 |
| EfficientBERT+ (Dong et al., 2021) | one-shot | 15.7M | 62ms | 89.3 | **89.9** | 92.4 | 38.1 | 85.1 | **83.0**/82.3 | 69.4 | **71.2** | 77.9 |
| EfficientBERT++ (Dong et al., 2021) | one-shot | 16.0M | 65ms | **90.6** | 88.9 | 92.3 | 42.5 | 83.6 | **83.0/82.5** | 67.8 | **71.2** | 78.0 |
| Synaptic Saliency (Abdelfattah et al., 2020) | zero-shot | 15.7M | 58ms | 89.4 | 88.1 | 91.0 | 33.6 | 83.1 | 82.6/81.1 | 70.6 | 70.3 | 76.6 |
| Activation Distance (Mellor et al., 2021a) | zero-shot | 15.6M | 60ms | 88.9 | 87.6 | 91.2 | 30.7 | 82.9 | 81.1/80.4 | 70.4 | 70.1 | 75.9 |
| Synaptic Diversity (Zhou et al., 2022) | zero-shot | 15.6M | 57ms | 88.3 | 88.1 | 91.5 | 25.8 | 84.7 | 81.3/80.2 | 70.6 | 70.3 | 75.6 |
| Head Confidence (Serianni & Kalita, 2023) | zero-shot | 15.6M | 63ms | 89.5 | 88.3 | 92.4 | 31.7 | 85.7 | 82.8/81.9 | **74.0** | 70.9 | 77.5 |
| Softmax Confidence (Serianni & Kalita, 2023) | zero-shot | 15.6M | 61ms | 88.4 | 87.5 | 90.8 | 32.5 | 83.5 | 81.2/80.5 | 70.3 | 69.9 | 76.1 |
| W-PCA-Tiny | zero-shot | 9.6M | 38ms | 88.7 | 87.6 | 91.9 | 27.4 | 84.8 | 81.1/79.8 | 71.1 | 70.3 | 75.9 |
| W-PCA-Small | zero-shot | 15.6M | 54ms | 90.3 | 88.7 | 91.5 | 38.4 | **86.4** | 82.8/82.2 | 73.8 | 70.8 | **78.3** |

## 6.2 IMPLEMENTATION DETAILS

### 6.2.1 SEARCH SPACE

The search space is illustrated in Figure 4, wherein we set the values of $m$ and $n$ to 12 and 6, respectively. Each block possesses a hidden size of 528, with the inner hidden size of the MobileBERT series blocks being one-fourth of the total hidden size. The hidden dimensions of the FFN increase by a factor of 132 for each multiple ranging from 1 to $n$. To calculate the value of PCA_dim, we set $\eta$ to 0.99, with the selection process detailed in Appendix D. To identify the combination of blocks that yields the highest PCA value, we utilize a genetic algorithm, the detailed implementation of which can be found in Appendix E. This algorithm uses a population size of 50 and a generation count of 40. The crossover probability is set to 1, the mutation probability to 0.1, and the upper limit for the model parameters is set to 15.7M, resulting in the W-PCA-Small model. By further reducing the upper limit for the model parameters to 10M and halving the number of layers ($m$), we obtain the W-PCA-Tiny model.

### 6.2.2 TRAINING

Once we obtain the desired architecture, we pretrain the model using the complete English Wikipedia (Devlin et al., 2019) and BooksCorpus (Zhu et al., 2015). We then proceed to fine-tune the model on each individual downstream task. During pretraining, the network is trained with a batch size set to 256. For the fine-tuning phase of the downstream tasks, the network is trained with a batch size set to 32. The CoLA task is trained for 50 epochs, while the other tasks are trained for 10 epochs. The learning rate is set at 0.0001 during pretraining. In the fine-tuning phase, the learning rate is set at 0.00005 for GLUE tasks and 0.0001 for SQuAD tasks. The training process utilizes the Adam optimizer with $\beta_1$ and $\beta_2$ values set at 0.9 and 0.999, respectively. The weight decay is set to 0.01. The learning rate decays linearly with a warm-up ratio set to 0.1. The KD loss function used in our approach is described in Appendix F.

## 6.3 RESULTS ON GLUE

### 6.3.1 MODEL ACCURACY AND LATENCY

Table 2 presents the results of the GLUE scores and model latency for the KD-based methods. Among them, except for the BERT-base teacher model we used ourselves, the results of all the manual and one-shot methods in the table are from relevant papers. Since zero-shot NAS methods have not been used in NLU tasks before, we applied the recent top-performing zero-shot proxy approaches on Transformer language models to the search space shown in Figure 4.

As shown in Table 2, under the search space depicted in Figure 4, our W-PCA metric achieved higher average scores on the GLUE test set compared to all baseline manual and one-shot methods. At the same time, it outperformed the previous state-of-the-art (SOTA) method EfficientBERT (Dong et al., 2021) in terms of parameter count, latency, and average score in the field of lightweight models.

Table 3: Comparison of results on the GLUE dev set with other NAS methods. The "Time" column represents the GPU days consumed by the NAS method search. It is not feasible to make a subcomparison with AutoBERT-Zero-small as it does not provide individual scores for each task in the GLUE dev set.

| Model | #Params | Time | QNLI | MRPC | SST-2 | CoLA | STS-B | MNLI-m | RTE | QQP | AVG |
|---|---|---|---|---|---|---|---|---|---|---|---|
| NAS-BERT-10 (Xu et al., 2021) | 10.0M | 96 d | 86.3 | 79.1 | 88.6 | 34.0 | 84.8 | 76.4 | 66.6 | 88.5 | 75.5 |
| NAS-BERT-30 (Xu et al., 2021) | 30.0M | 96 d | 88.4 | 84.6 | 90.5 | 48.7 | **87.6** | 81.0 | 71.8 | **90.2** | 80.3 |
| EfficientBERT-TINY (Dong et al., 2021) | 9.4M | 58 d | 89.3 | 90.1 | 90.1 | 39.1 | 79.9 | 81.7 | 63.2 | 86.7 | 77.5 |
| EfficientBERT (Dong et al., 2021) | 15.7M | 58 d | 90.4 | **91.5** | 91.3 | **50.2** | 82.5 | **83.1** | 66.8 | 87.3 | 80.4 |
| AutoBERT-Zero-small (Gao et al., 2022) | 13.0M | ~1,000 d | - | - | - | - | - | - | - | - | 80.5 |
| Synaptic Diversity (Zhou et al., 2022) | 15.6M | 0.7 d | 88.9 | 87.6 | 91.4 | 32.0 | 84.1 | 81.0 | 73.4 | 88.2 | 78.3 |
| Head Confidence (Serianni & Kalita, 2023) | 15.6M | 0.5 d | 90.1 | 89.7 | 92.4 | 37.5 | 84.1 | 82.5 | 75.9 | 89.1 | 80.2 |
| Softmax Confidence (Serianni & Kalita, 2023) | 15.6M | 0.5 d | 89.4 | 88.3 | 92.0 | 32.6 | 84.7 | 81.6 | 73.9 | 88.9 | 78.9 |
| W-PCA-Tiny | 9.6M | 0.4 d | 89.2 | 89.2 | 92.0 | 33.2 | 84.0 | 80.5 | 71.1 | 88.0 | 78.4 |
| W-PCA-Small | 15.6M | 0.5 d | **90.8** | 90.5 | **92.8** | 44.0 | 85.3 | 82.9 | **76.1** | 88.8 | **81.4** |

Additionally, W-PCA achieved the highest score on the STS-B task. It is worth noting that, in the same search space, the optimal structure found by W-PCA surpasses all previous zero-shot methods (Abdelfattah et al., 2020; Mellor et al., 2021a; Zhou et al., 2022; Serianni & Kalita, 2023) applied to Transformer language models, highlighting its exceptional ability in exploring optimal network structures in zero-shot NAS methods.

### 6.3.2 SEARCH EFFICIENCY

As shown in Table 3, under our search space, the search efficiency of all zero-shot proxies (including our W-PCA method) has been improved by two to three orders of magnitude compared to previous training-based NAS, and achieved competitive performance. The three zero-shot proxies, Synaptic Diversity (Zhou et al., 2022), Head Confidence (Serianni & Kalita, 2023), and Softmax Confidence (Serianni & Kalita, 2023), can compete with the optimal structures found by previous training-based NAS in our search space. Our W-PCA method surpasses all previous training-based methods in the field of lightweight language models in terms of average score and achieves the best average score. Moreover, in three out of eight tasks, W-PCA achieves the highest performance. Our method discovers the latest SOTA effects in the field of lightweight models with almost negligible search cost, reducing greenhouse gas $CO_2$ emissions by two to three orders of magnitude [1], and significantly improving the utilization of global energy resources.

It is also worth noting that in the internal comparison of zero-shot proxies, Head Confidence (Serianni & Kalita, 2023), Softmax Confidence (Serianni & Kalita, 2023), and our W-PCA method require shorter search time than the Synaptic Diversity (Zhou et al., 2022) method, which needs to compute gradients, by an additional 0.2 GPU days. Additionally, our W-PCA-Tiny model has a lower parameter limit set during the search, resulting in slightly lower computation time for the forward propagation of each neural network individual, thus reducing the search time by 0.1 GPU days compared to the W-PCA-Small model.

### 6.4 RESULTS ON SQUAD

Table 4: Results on SQuAD dev sets. [*]: our implementation.

| Model | #Params | SQuAD v1.1 EM/F1 | SQuAD v2.0 EM/F1 |
|---|---|---|---|
| BERT-base | 108.9M | 80.8/88.5 | -/- |
| BERT-base[*] | 108.9M | 80.7/88.2 | 75.7/78.7 |
| TinyBERT-4 | 14.5M | 72.7/82.1 | 68.2/71.8 |
| MiniLM-6 | 22.9M | -/- | -/72.7 |
| EfficientBERT++ | 16.0M | 78.3/86.5 | 73.0/76.1 |
| W-PCA-Tiny | 9.6M | 74.6/83.5 | 69.0/72.1 |
| W-PCA-Small | 15.6M | **78.4/86.7** | **73.3/76.8** |

We compared the W-PCA proposed in this article with manually designed lightweight models, namely TinyBERT (Jiao et al., 2020), MiniLM (Wang et al., 2020), and the one-shot NAS method EfficientBERT (Dong et al., 2021), on the SQuAD dataset. The results are presented in Table 4. Despite having fewer parameters than TinyBERT-4, MiniLM-6, and Efficient++, the W-PCA-Small model outperforms these methods in terms of both EM and F1 scores on both the SQuAD v1.1 and SQuAD v2.0 datasets. This observation demonstrates the robust adaptability of the investigated models across diverse datasets.

---

[1]A reduction of 0.34k lbs and 3.4k lbs of $CO_2$ emissions for every 100-fold and 1,000-fold, respectively (Strubell et al., 2019).

Table 5: Comparison results of W-PCA and its product counterparts as proxies on the GLUE dev set.

| Proxy | #Params | QNLI | MRPC | SST-2 | CoLA | STS-B | MNLI-m | RTE | QQP | AVG |
|-------|---------|------|------|-------|------|-------|--------|-----|-----|-----|
| #Params | 15.7M | 89.3 | 88.8 | 90.7 | 43.8 | 83.6 | 82.6 | 76.1 | 87.5 | 80.3 |
| V-PCA | 15.6M | 89.9 | 91.4 | 92.7 | 39.4 | 84.9 | 82.9 | 76.0 | 88.9 | 80.8 |
| W-PCA | 15.6M | 90.8 | 90.5 | 92.8 | 44.0 | 85.3 | 82.9 | 76.1 | 88.8 | 81.4 |

## 6.5 ABLATIONS

In order to investigate the effects of each component of W-PCA on the experimental results, we performed ablation experiments. Specifically, we utilized the components of W-PCA, as described in Equation (8) where the first component is the number of parameters (#Params) and the second component is the V-PCA value (defined in Equation (7)), as fitness values for the genetic algorithm to explore the optimal network structure in Section 6.2.1. We then compared the performance of the discovered network structures with W-PCA.

The results, presented in Table 5, demonstrate that by multiplying the number of parameters with the V-PCA value and using W-PCA as the zero-shot evaluation metric, the performance of the searched networks significantly improves compared to using either #Params or V-PCA alone as the evaluation metric.

Encouragingly, the incorporation of an extra feature does not necessitate a significant rise in computational time, thereby rendering the multiplication approach highly efficient. For further ablations, please refer to Appendix H.

## 7 CONCLUSION

In this paper, we propose W-PCA [2], and significantly improving the utilization of global energy resources., a novel zero-shot NAS method specifically designed for lightweight language models. In the ranking correlation experiments conducted on the search space of FlexiBERT, W-PCA achieves a Kendall $\tau$ score that surpasses the previous method by 0.220 and a Spearman $\rho$ score that surpasses the previous method by 0.334. In the accuracy experiments conducted on GLUE and SQuAD, W-PCA not only achieves the highest score, but also significantly improves search efficiency. On the GLUE test set, W-PCA improves search efficiency by over a hundredfold compared to the previous best-performing one-shot NAS method, with an average score improvement of 0.3. On the GLUE dev set, W-PCA improves search efficiency by 2,000 times and achieves an average score improvement of 0.9 compared to the previous best-performing one-shot NAS method. In future work, we will extend our approach to more compression tasks (Dong et al., 2024a; Li et al., 2024b;e; Dong et al., 2024b). Our work contributes to the advancement of NAS methods for lightweight language models, enabling the design and optimization of efficient and effective systems for natural language processing.

**Limitations** This work focuses on the ranking correlation tasks commonly addressed in prior zero-shot NAS methods and on NLU tasks relevant to lightweight models. However, recent language model research increasingly centers on generative models with over 1B parameters. In Appendix I, we further discuss potential extensions related to these large-scale generative models.

### ACKNOWLEDGMENTS

Special thanks to Professor Yajun Ha from ShanghaiTech University. During the resubmission of this paper to ICLR, he helped us summarize the contributions of the paper and suggested a more suitable title.

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

Table 6: Performance comparison on the GLUE test set for a model composed of 12 identical blocks.

| Block Type | #Params | QNLI | MRPC | SST-2 | CoLA | STS-B | MNLI-m/mm | RTE | QQP | AVG |
|---|---|---|---|---|---|---|---|---|---|---|
| BERT (Devlin et al., 2019) | 24.8M | 90.9 | 89.0 | 92.9 | 46.7 | 87.2 | 84.4/83.5 | 71.0 | 71.9 | 79.7 |
| MobileBERT (Sun et al., 2020) | 10.4M | 87.2 | 87.7 | 91.4 | 31.8 | 84.9 | 81.5/81.4 | 69.6 | 70.8 | 76.3 |
| LiteTransformer (Wu et al., 2019) | 34.6M | 90.2 | 89.2 | 92.5 | 36.4 | 86.4 | 83.1/82.3 | 74.0 | 70.8 | 78.3 |
| ConvBERT (Jiang et al., 2020) | 37.4M | 89.7 | 89.4 | 91.9 | 36.0 | 85.8 | 82.5/82.0 | 72.1 | 70.7 | 77.8 |

## A  WHY BERT AND MOBILEBERT?

As shown in Table 6, we have attempted to train a model by composing 12 blocks of the same type, with a dimension of 528 for FFN. The results indicate that BERT and MobileBERT blocks outperform other blocks in terms of the unit parameter performance. Therefore, we have included these two fundamental blocks in our search space. In fact, if we construct a supernet as described in Section H.3, the optimal network structure selected by the genetic algorithm will also only choose BERT and MobileBERT blocks, without considering any other types of blocks.

## B  TRAINING DETAILS OF THE FLEXIBERT SEARCH SPACE

Table 7: The FlexiBERT search space comprises a total of 10,621,440 architectures.

| Architecture Element | | Hyperparameters Values |
|---|---|---|
| Embedding dimension | | {128, 256} |
| Number of Encoder Layers | | {2, 4} |
| Type of attention operator | | {self-attention, linear transform, span-based dynamic convolution} |
| Number of operation heads | | {2, 4} |
| Hidden dimension | | {512, 1024} |
| Number of feed-forward stacks | | {1, 3} |
| Attention operation | if self-attention | {scaled dot-product, multiplicative} |
| | if linear transform | {discrete Fourier, discrete cosine} |
| | if dynamic convolution | convolution kernel size: {5, 9} |

All transformer architectures within the search space were trained on TPUv2s with 8 cores and 64 GB of memory using Google Colaboratory. The entire process of pretraining and finetuning the benchmark took approximately 25 TPU days. For the evaluation of training-free metrics, 2.8 GHz Intel Cascade Lake processors with either 16 or 32 cores and 32 GB of memory were employed.

In terms of hyperparameter settings, except for setting the training steps to 100,000 during the pre-training phase, everything else is the same as training ELECTRA-Small. Specifically, during the pre-training phase, the generator size multiplier is set to 1/4, the mask percentage is set to 15%, the warmup step is 10,000, the learning rate is 5e-4, and the batch_size is 128. During the fine-tuning phase, the learning rate is 3e-4, the layerwise $lr$ decay is 0.8, the warmup fraction is 0.1, the attention dropout is 0.1, and the batch_size is 32. For the RTE and STS tasks, 10 epochs are trained, while for other tasks, 3 epochs are trained. Both during pre-training and fine-tuning phases, the learning rate decay is linear, the vocabulary size is 30522, the dropout is 0.1, the weight decay value is 0.01, the $\epsilon$ value for the Adam optimizer is 1e-6, $\beta_1$ value is 0.9, and $\beta_2$ value is 0.999.

## C  ABLATIONS OF RANKING EVALUATION

To examine the stability of zero-shot metrics, we conducted a series of studies to investigate the effects of random architecture initialization (Figure 5) and varying batch inputs (Figure 6) on the evaluation of zero-shot metrics within the FlexiBERT search space. The range of fluctuations observed in various zero-shot metrics in a neural network model demonstrates that, while the number of parameters remains unaffected, other zero-shot metrics exhibit varying degrees of fluctuation depending on the initialization of parameters and batch inputs. The figure indicates that PCA alone, serving as a proxy metric, shows good stability, and W-PCA exhibits the same level of stability as PCA due to multiplication by a constant parameter that does not alter the magnitude of the original

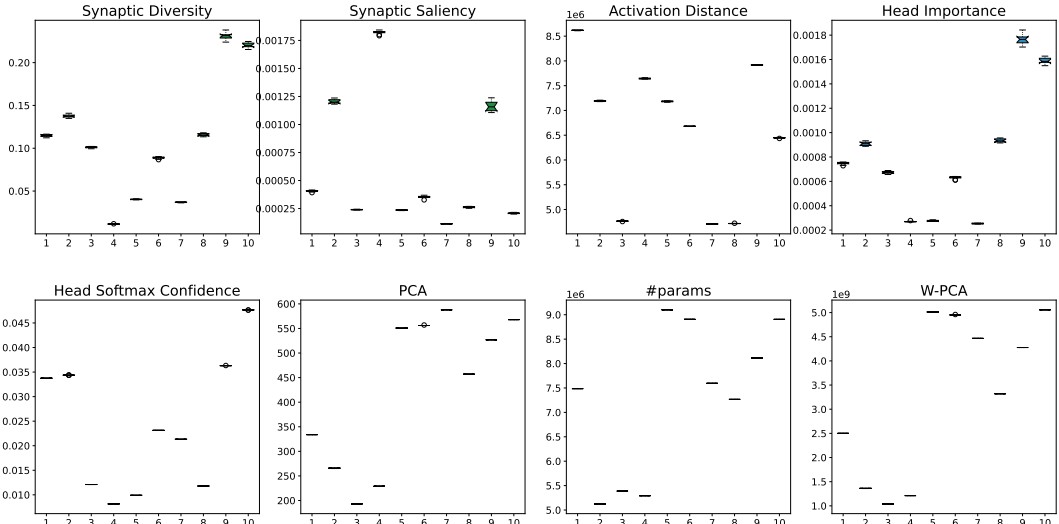

Figure 5: Evaluation of zero-shot metrics with various initialization weights in the FlexiBERT search space. Ten architectures are randomly sampled from the search space, representing decile ranges of the GLUE score (e.g., 0-10%, 10-20%, ..., 90-100%). To ensure robustness, ten different random seeds are employed for weight initialization.

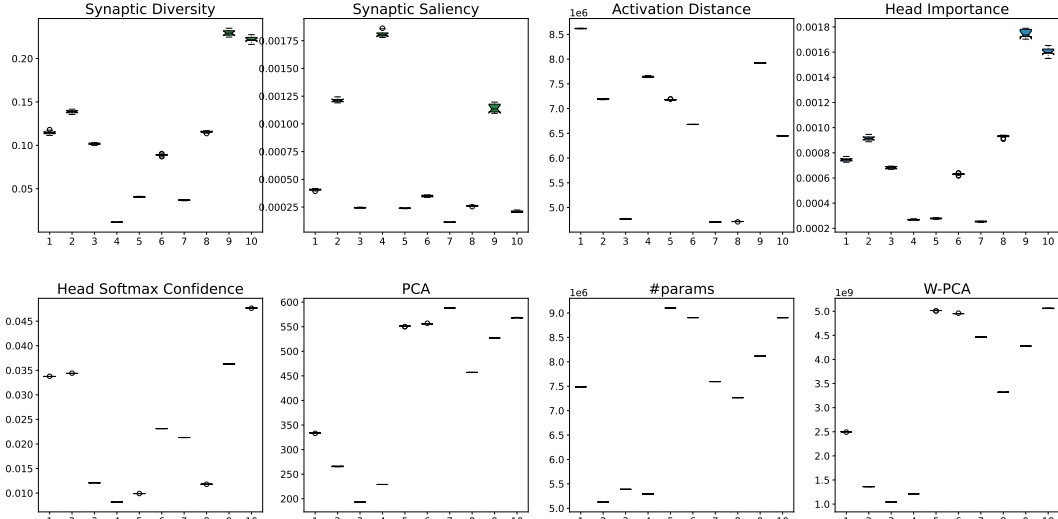

Figure 6: Evaluation of zero-shot metrics with various minibatch inputs in the FlexiBERT search space. Ten architectures are randomly sampled from the search space, representing decile ranges of the GLUE score (e.g., 0-10%, 10-20%, ..., 90-100%). The same ten minibatches, each with a size of 128, are randomly chosen from the OpenWebText dataset for each architecture and metric.

proxy metric's fluctuation. When compared to other zero-shot metrics, PCA and W-PCA exhibit slightly better stability than Activation Distance (Mellor et al., 2021a) and Head Softmax Confidence (Serianni & Kalita, 2023), and significantly better stability than Synaptic Diversity (Zhou et al., 2022), Synaptic Saliency (Abdelfattah et al., 2020), and Head Importance (Serianni & Kalita, 2023). These findings demonstrate that our proposed method not only achieves superior ranking evaluation but also demonstrates improved stability.

# D    ROBUSTNESS WITH DIFFERENT $\eta$'S

Table 8: Principal component dimensions at different $\eta$ values for a neural network model composed of 12 identical blocks during training.

| Model | $\eta = 0.9$ | $\eta = 0.99$ | $\eta = 0.999$ | $\eta = 1$ |
|---|---|---|---|---|
| BERT | 91 (17.2%) | 159 (30.1%) | 286 (54.2%) | 528 |
| MobileBERT | 58 (43.9%) | 101 (76.5%) | 125 (94.7%) | 132 |

In our NLU task experiments, when training networks composed of 12 identical BERT or MobileBERT blocks, we observed an extremely uneven distribution of principal components. Specifically, the largest eigenvalues, when sorted, are predominantly concentrated at the beginning, while the remaining eigenvalues are close to zero. The dimensions required for each layer to achieve different $\eta$ values when training these models are shown in Table 8. From the table, it is evident that the BERT blocks exhibit a more uneven distribution compared to the MobileBERT blocks, as only 54.2% of the dimensions are needed to reach a 0.999 principal component contribution rate. However, setting $\eta$ to 0.999 would result in a loss of distinction for MobileBERT blocks, as they would only have integer values between 125 and 132. Since our search space includes networks composed of both BERT and MobileBERT blocks, we compromised by setting $\eta$ to 0.99.

Table 9: Results obtained from varying $\eta$ values in rank correlation experiments.

| Proxy | $\eta = 0.9$ | | $\eta = 0.99$ | | $\eta = 0.999$ | |
|---|---|---|---|---|---|---|
| | $\tau$ | $\rho$ | $\tau$ | $\rho$ | $\tau$ | $\rho$ |
| Vanilla PCA | 0.466 | 0.677 | 0.449 | 0.667 | 0.433 | 0.653 |
| W-PCA | 0.526 | 0.698 | 0.513 | 0.689 | 0.499 | 0.688 |

For the ranking correlation experiments, each block has more combination possibilities. We adjusted different $\eta$ values and obtained the following ranking correlations in Table 9. As shown in the table, both V-PCA and W-PCA maintain high rank correlation regardless of the $\eta$ value, demonstrating the robustness of the $\eta$ variable in the rank correlation experiments. Additionally, W-PCA consistently outperforms V-PCA in both Kendall's $\tau$ and Spearman's $\rho$ correlations across different $\eta$ values, further indicating that incorporating the number of parameters as a factor in the proxy enhances the stability of the rank correlation results.

# E    IMPLEMENTATION DETAILS OF GENETIC ALGORITHM

The genetic algorithm is a widely used classic algorithm for solving combinatorial optimization problems. In Figure 7, we illustrate the flowchart of the genetic algorithm. Early NAS algorithms (Real et al., 2019; Sun et al., 2019) often encoded each individual neural network and then trained them from scratch. To efficiently identify the best combination of lightweight BERT blocks, we encoded and solved the combination using a genetic algorithm.

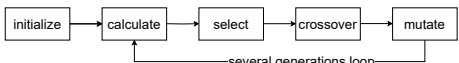

## E.1    ENCODE

When there are $m$ layers in the combination, we use an integer array of length $m$ to encode the selection method for each layer. Each integer in the array represents the type of block chosen for that layer. Taking the 12-layer model of W-PCA-Small as an example, if the number at a given position is less than or equal to 5, it indicates the selection of a BERT-type block; otherwise, a MobileBERT-type block is chosen. Each integer represents the dimension of the hidden dimension. Let $x$ be the integer at the current layer position, then the hidden dimension is calculated as $132 \times (x\%6 + 1)$.

Figure 7: Diagram of genetic algorithm. In order to effectively solve practical problems using a genetic algorithm, it is crucial to define a suitable method for encoding the solutions. Additionally, in each generation, the fitness of every individual should be calculated, and only the most excellent individuals should be selected for the application of crossover and mutation operators. These operators are used to generate the individuals that will make up the next generation.

## E.2 CROSSOVER

---

**Algorithm 1:** Crossover operation in genetic algorithm

---

**Input:** Parent 1 encoding $p1$, Parent 2 encoding $p2$
**Output:** Offspring encoding

1 **Function** Crossover($p1, p2$):
2     Create an empty offspring encoding $child$
3     **for** $i \leftarrow 1$ **to** $length(p1)$ **do**
4         **if** *random number* < *0.5* **then**
5             Add the gene from the corresponding position in the parent 1 encoding to the offspring encoding
6             $child[i] \leftarrow p1[i]$
7         **else**
8             Add the gene from the corresponding position in the parent 2 encoding to the offspring encoding
9             $child[i] \leftarrow p2[i]$
10         **end**
11     **end**
12     **return** $child$

---

We employ the single-point crossover method as outlined in Algorithm 1 to produce offspring encodings. In each generation, parents are randomly chosen from the top 10 individuals, and offspring are generated through crossover until the desired number of offspring is obtained.

## E.3 MUTATION

We set the mutation probability to 0.1. When the current position mutation is triggered, the current position will randomly mutate into another integer within the selection range.

## F KD LOSS FUNCTION

The distillation loss function of EfficientBERT (Dong et al., 2021) forms the basis of our approach. For the student model, we define $\mathcal{L}_{attn}^{i}$ as the loss for the multi-head attention (MHA) output and $\mathcal{L}_{hidd}^{i}$ as the loss for the feed-forward network (FFN) output in the $m$-th layer. The embedding loss, represented by $\mathcal{L}_{embd}$, is also included. These losses are calculated using the mean squared error (MSE) as follows:

$$\begin{cases} \mathcal{L}_{attn}^{i} = \mathrm{MSE}(\mathbf{A}_i^S \mathbf{W}_a, \mathbf{A}_j^T), \\ \mathcal{L}_{hidd}^{i} = \mathrm{MSE}(\mathbf{H}_i^S \mathbf{W}_h, \mathbf{H}_j^T), \\ \mathcal{L}_{embd} = \mathrm{MSE}(\mathbf{E}^S \mathbf{W}_e, \mathbf{E}^T) \end{cases} \tag{9}$$

Here, $\mathbf{A}_i^S$ and $\mathbf{H}_i^S$ represent the outputs of the MHA and FFN layers, respectively, in the $i$-th layer of the student model. Similarly, $\mathbf{A}_j^T$ and $\mathbf{H}_j^T$ represent the outputs of the MHA and FFN layers, respectively, in the $j$-th layer of the teacher model corresponding to the $i$-th layer of the student model.

For our fixed teacher model, BERT-base, which comprises 12 layers, a one-to-one sequential correspondence exists between the layers of the student and teacher models when both models have 12 layers. However, in the case of a student model with only 6 layers, the correspondence remains one-to-one, but with a 2-layer interval. This implies that the first layer of the student model corresponds to the second layer of the teacher model, and so forth, until the sixth layer of the student model aligns with the twelfth layer of the teacher model.

The trainable matrices $\mathbf{W}_a$, $\mathbf{W}_h$, and $\mathbf{W}_e$ are used to adjust the dimensionality of the student and teacher models. Additionally, we define $\mathcal{L}_{pred}$ as the prediction loss, which is calculated using soft cross-entropy (CE):

$$\mathcal{L}_{pred} = \mathrm{CE}(\mathbf{z}^S, \mathbf{z}^T) \tag{10}$$

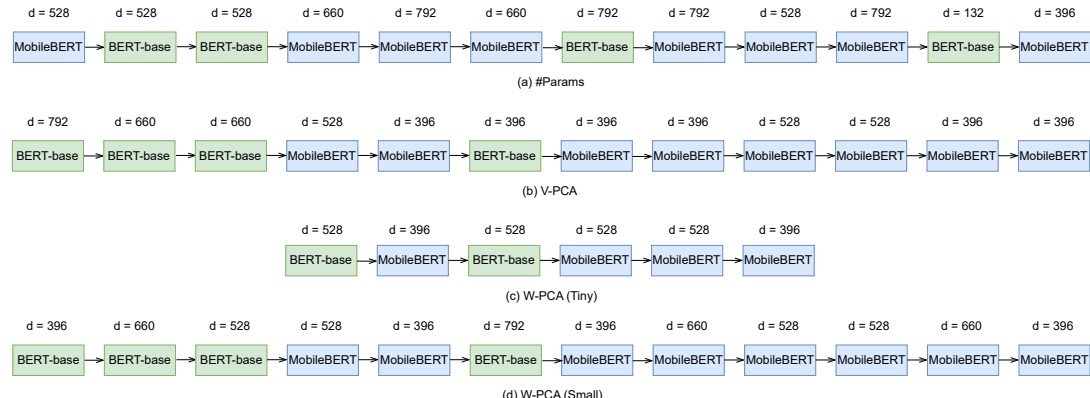

Figure 8: Visualizations of the searched architectures, where d represents the hidden dimensions.

Here, **z** represents the predicted logit vector.

The total loss is a combination of the above terms:

$$\mathcal{L} = \sum_{i=1}^{m}(\mathcal{L}_{attn}^{i} + \mathcal{L}_{hidd}^{i}) + \mathcal{L}_{embd} + \gamma\mathcal{L}_{pred} \tag{11}$$

The coefficient $\gamma$ is used to control the contribution of the predicted loss. It is set at 0 during the pretraining phase and 1 during the fine-tuning phase.

## G VISUALIZATION OF ARCHITECTURES

Figure 8 illustrates the schematic diagram of the network structure. It is observed that all models preferentially choose MobileBERT as the candidate block, suggesting that MobileBERT is better suited for lightweight language models in comparison to BERT-base. Furthermore, with the exception of the searched model that solely relies on parameter count as the search evaluation metric, the candidate blocks of MobileBERT are predominantly located in the higher layers, indicating that this architecture is more adept at analyzing high-level semantic information.

## H MORE ABLATIONS ON ACCURACY COMPARISION

### H.1 DIFFERENT INITIALIZATIONS

Table 10: Statistical significance tests based on average scores from 3 runs with different random seed initializations on the GLUE dev set.

| Proxy | #Params | V-PCA | W-PCA |
|---|---|---|---|
| **GLUE** | 80.3±0.0 | 80.9±0.09 | 81.5±0.14 |

To validate the stability of each component of W-PCA, we conducted multiple runs using #Params, V-PCA, and W-PCA as proxies. The results are presented in Table 10. As shown in the table, under different weight initializations, the structures identified using #Params as a proxy remain identical across runs since #Params does not change. However, when V-PCA and W-PCA are used as proxies, the genetic algorithm identifies different structures, resulting in varying GLUE scores after training. On average, W-PCA demonstrates a clear advantage over V-PCA, and both W-PCA and V-PCA outperform #Params. This further validates the effectiveness of the proposed proxy.

Table 11: Performance comparison of larger-scale models on the GLUE test set.

| Model | #Params | QNLI | MRPC | SST-2 | CoLA | STS-B | MNLI-m/mm | RTE | QQP | AVG |
|---|---|---|---|---|---|---|---|---|---|---|
| TinyBERT-6 (Jiao et al., 2020) | 67.0M | 89.8 | 89.0 | 92.0 | 38.8 | 83.1 | 83.8/83.2 | 65.8 | 71.4 | 77.4 |
| EfficientBERT (Dong et al., 2021) | 70.1M | 90.4 | 89.0 | 92.6 | 46.2 | 83.7 | 84.1/83.2 | 67.7 | 74.4 | 78.7 |
| W-PCA-Large | 66.9M | 90.9 | 88.7 | 93.0 | 40.0 | 87.5 | 84.6/83.3 | 75.6 | 71.5 | 79.5 |

## H.2 COMPARISON OF LARGER-SIZED MODELS

In the main body of the paper, we primarily focus on lightweight language models with parameter sizes of approximately 15M and 10M. To investigate the applicability of W-PCA on larger language models in search processes, we carried out experiments by utilizing a larger model that expanded the size of the search space as described in Section 6.2.1. Specifically, we doubled the hidden_size and the hidden_dimension of $n$ candidate dimensions. Moreover, we increased the parameter limit in the genetic algorithm to 67M, resulting in the creation of our W-PCA-Large model. Our model, presented in Table 11, outperforms TinyBERT-6 (Jiao et al., 2020) and EfficientBERT (Dong et al., 2021) models of similar scale in terms of average GLUE score, despite having a slightly lower parameter count. This indicates that W-PCA also exhibits strong adaptability in larger search spaces.

## H.3 COMPARISON WITH ONE-SHOT NAS

Table 12: Comparison of search and training stages for one-shot and zero-shot methods.

| Method | Search Stage | | | Training Stage | |
|---|---|---|---|---|---|
| one-shot | Pretrain the supernet | Finetune the supernet for downstream tasks | Finding the optimal neural network structure using genetic algorithm | Re-pretrain the optimal network structure | Fine-tune for each downstream task |
| zero-shot | Finding the optimal neural network structure using genetic algorithm | | | | |

In Section 4, we refer to the composition method of the supernet in the one-shot NAS method SPOS (Guo et al., 2020), and construct the combination method of the lightweight BERT model in our zero-shot NAS search space. In order to compare its performance with the one-shot NAS, we now build a real supernet, as shown in Figure 4, with a total of $m$ layers and $2 \times n$ candidate blocks in each layer. Before searching for the optimal structure using a genetic algorithm, we performed one round of pre-training and fine-tuning based on the SPOS method. During each batch, a random path is selected from the $(2 \times n)^m$ combinations for forward propagation and backward parameter updates. After completing the pre-training and fine-tuning process, we proceeded with the workflow described in the zero-shot NAS experiment. This involved searching for the optimal network architecture on the supernet using a genetic algorithm, and then re-pretraining and fine-tuning this optimal architecture for downstream tasks.

**Implementation Details.** We first pretrain the supernet on English Wikipedia (Devlin et al., 2019) and BooksCorpus (Zhu et al., 2015), then utilize 90% of the training set from each GLUE task for fine-tuning. We reserve the remaining 10% of the MNLI task to evaluate the accuracy of architectures in the search. During the pre-training and fine-tuning process, the number of epochs is set to 10, and the batch_size is set to 256 for both. The learning rate for pre-training is set to 1e-4, and the learning rate for fine-tuning is set to 4e-4. The optimizer, weight decay, and learning rate adjustment strategy are the same as in the training section. The loss function used is still the MSE loss function described in Appendix F.

**Results & Analysis.** As displayed in Table 13, despite extensive computational resources devoted to the one-shot NAS search, the performance enhancement for various-sized models of W-PCA is not significant. To further investigate the reasons behind this, we outline the steps of one-shot NAS and zero-shot NAS in Table 12. Both approaches involve finding the optimal network structure through a search stage, followed by training this structure in the subsequent training stage. In the search stage of one-shot NAS, a supernet training is necessary, whereas zero-shot NAS only requires temporary construction of a neural network at each sampling step to compute the zero-shot proxy. Consequently, the memory overhead for zero-shot NAS is minimal. Contrasting with zero-shot NAS, one-shot NAS incurs additional time overhead due to pre-training the supernet on the corpus and performing global fine-tuning on downstream tasks. Additionally, one-shot NAS possesses extra pre-trained weights

Table 13: Comparison of zero-shot and one-shot methods on the GLUE test set in the same search space. "Time" also refers to the GPU time consumption in the NAS stage.

| Model | Type | #Params | Time | QNLI | MRPC | SST-2 | CoLA | STS-B | MNLI-m/mm | RTE | QQP | AVG |
|---|---|---|---|---|---|---|---|---|---|---|---|---|
| W-PCA-Tiny | zero-shot | 9.6M | 0.4 d | 88.7 | 87.6 | 91.9 | 27.4 | 84.8 | 81.1/79.8 | 71.1 | 70.3 | 75.9 |
| | one-shot | 9.7M | 24 d | 89.2 | 87.5 | 92.3 | 28.9 | 83.7 | 81.4/80.5 | 71.4 | 70.5 | 76.2 |
| W-PCA-Small | zero-shot | 15.6M | 0.5 d | 90.3 | 88.7 | 91.5 | 38.4 | 86.4 | 82.8/82.2 | 73.8 | 70.8 | 78.3 |
| | one-shot | 15.6M | 28 d | 90.3 | 88.9 | 92.5 | 36.1 | 86.7 | 83.7/82.5 | 74.4 | 70.6 | 78.4 |

Table 14: Comparison of zero-shot NAS methods on the GLUE dev set within the EfficientBERT search space.

| Proxy | #Params | Time | QNLI | MRPC | SST-2 | CoLA | STS-B | MNLI-m | RTE | QQP | AVG |
|---|---|---|---|---|---|---|---|---|---|---|---|
| Synaptic Diversity (Zhou et al., 2022) | 15.6M | 1.0 d | 84.1 | 85.2 | 86.1 | 21.1 | 76.5 | 77.2 | 66.5 | 78.2 | 71.9 |
| Head Confidence (Serianni & Kalita, 2023) | 15.7M | 0.8 d | 83.1 | 84.4 | 86.8 | 21.8 | 80.9 | 76.3 | 66.1 | 84.6 | 73.0 |
| Softmax Confidence (Serianni & Kalita, 2023) | 15.6M | 0.8 d | 84.0 | 83.0 | 87.3 | 21.2 | 78.7 | 76.7 | 68.3 | 77.6 | 72.1 |
| W-PCA | 15.6M | 0.8 d | 86.0 | 85.3 | 90.1 | 24.4 | 80.1 | 76.5 | 65.8 | 86.2 | 74.3 |

apart from the different searched network structures. After the network structure is searched by zero-shot NAS, no pre-training weights are used, and the weights are randomly initialized, resulting in relatively minor effects on accuracy. Therefore, zero-shot NAS remains the most cost-effective search solution.

## H.4 EXPERIMENTS ON EFFICIENTBERT SEARCH SPACE

Previous zero-shot NAS studies have primarily assessed performance on NAS Benchmark datasets, using rank correlation to evaluate the effectiveness of these zero-shot proxies. To facilitate a comparison with other zero-shot proxies applied to Transformer models in text tasks, we designed the search space described in Section 4 and compared our method within this context. To further ensure robustness, we conducted additional comparisons using the EfficientBERT search space, with the results shown in Table 14. The data demonstrate that our proposed zero-shot proxy maintains a significant advantage in accuracy across various search spaces.

## I   CAUSAL LANGUAGE MODELING TASK

To validate our performance more sufficiently, we further transfer our method to causal language modeling (CLM) task.

### I.1   SETUP

**Searching.** We use some 1B~3B parameter OPT  (Zhang et al., 2022) and LLaMA3  (Dubey et al., 2024) models as baseline models, keeping the number of layers unchanged while searching for the optimal combinations of MHA blocks and FFN dimensions in each layer of the models. For a fair comparison, we set the maximum parameter limit for the searched models to be the same as that of the baseline models.

**Pretraining.** We pretrain the models for causal language modeling (CLM) using a randomly selected 1% of texts from the SlimPajama dataset  (Soboleva et al., 2023). For this task, we train the models generated by each zero-shot proxy on 8 NVIDIA V100 GPUs, with a total batch size of 64 for 3 epochs. A cosine annealing schedule is applied to the learning rate, starting at $5 \times 10^{-4}$. Optimization is performed using the AdamW optimizer with a weight decay of 0.1. To accommodate the differences in configurations, such as the tokenizer and training loss, between OPT and LLaMA3, we train two separate models for each proxy.

**Instruction finetuning.** To obtain models suitable for chat applications, we finetune each pretrained model using the instruction dataset databricks-dolly-15k  (Conover et al., 2023) with the KD method MiniLLM  (Gu et al., 2024). Specifically, we use OPT-13B and LLaMA3.2-11B as the teacher model to supervise other models. The training data and strategies employed are consistent with those of MiniLLM.

Table 15: Evaluation results referring to the average GPT-4 feedback scores (GPT4) and Rouge-L scores (R-L) obtained from 5 runs.

| Teacher | Proxy | #Params | DollyEval GPT4 | DollyEval R-L | SelfInst GPT4 | SelfInst R-L | VicunaEval GPT4 | VicunaEval R-L | S-NI R-L | UnNI R-L |
|---|---|---|---|---|---|---|---|---|---|---|
| OPT-13B | OPT-1.3B (baseline) | - | 60.7 | 26.7 | 47.0 | 14.8 | 50.6 | 17.9 | 28.6 | 33.4 |
| | Synaptic Diversity (Zhou et al., 2022) | 1.3B | 60.4 | 26.9 | 47.3 | 15.2 | 51.8 | 17.6 | 29.1 | 34.2 |
| | Head Confidence (Serianni & Kalita, 2023) | 1.3B | 61.5 | 27.3 | **48.0** | 15.3 | 51.5 | 18.1 | **29.5** | 34.3 |
| | Softmax Confidence (Serianni & Kalita, 2023) | 1.3B | 61.2 | 27.1 | 47.5 | 14.6 | 50.8 | 18.2 | 28.9 | 33.9 |
| | W-PCA | 1.3B | **62.0** | **27.6** | 47.9 | **16.1** | **52.3** | **19.0** | 29.2 | **35.0** |
| OPT-13B | OPT-2.7B (baseline) | - | 63.2 | 27.4 | 52.7 | 17.2 | 55.9 | 19.1 | 30.7 | 35.1 |
| | Synaptic Diversity (Zhou et al., 2022) | 2.7B | 63.8 | 27.5 | 52.9 | 17.8 | 56.2 | 18.9 | 31.5 | 35.8 |
| | Head Confidence (Serianni & Kalita, 2023) | 2.7B | 64.0 | 27.8 | 53.2 | 17.5 | 56.5 | 19.4 | 31.6 | 35.9 |
| | Softmax Confidence (Serianni & Kalita, 2023) | 2.7B | 63.2 | 27.8 | 53.0 | 17.3 | 56.1 | 19.2 | 31.0 | 35.4 |
| | W-PCA | 2.7B | **64.5** | **28.3** | **53.9** | **18.1** | **57.2** | **20.1** | **32.1** | **36.2** |
| LLaMA3.2-11B | LLaMA3.2-3B (baseline) | - | 74.0 | 29.8 | 69.1 | 23.6 | 64.6 | 25.2 | 36.2 | 43.9 |
| | Synaptic Diversity (Zhou et al., 2022) | 3B | 73.8 | 30.5 | 70.1 | 23.8 | 65.0 | 25.4 | 36.7 | 44.8 |
| | Head Confidence (Serianni & Kalita, 2023) | 3B | 74.4 | 30.9 | 71.4 | 24.2 | 65.2 | 26.2 | 37.1 | **45.6** |
| | Softmax Confidence (Serianni & Kalita, 2023) | 3B | 74.1 | 30.1 | 70.4 | 23.6 | 64.9 | 25.9 | 37.2 | 44.3 |
| | W-PCA | 3B | **75.1** | **31.4** | **71.8** | **24.7** | **65.9** | **26.8** | **37.6** | 45.2 |

Each model required approximately 20 GPU days for pretraining and around 28 GPU hours for finetuning in our experiments.

**Quantitative evaluation.** We perform a numerical evaluation on five instruction-following datasets:

- **DollyEval**: This is a 500-sample test set that we extracted from the databricks-dolly-15k dataset.
- **SelfInst** (Wang et al., 2022a): A user-oriented instruction-following set comprising 252 samples.
- **VicunaEval** (Chiang et al., 2023): The evaluation includes 80 challenging questions used in the Vicuna project.
- **S-NI**: The test set of Super-NaturalInstructions (Wang et al., 2022b), which consists of 9,000 samples across 119 tasks.
- **UnNI**: The core set of UnnaturalInstructions (Honovich et al., 2022), which contains 60,000 samples.

## I.2 QUANTITATIVE RESULTS ON LANGUAGE BENCHMARKS

We evaluated the performance of the W-PCA model by comparing it with other models trained using MiniLLM. As shown in Table 15, across various model sizes, W-PCA consistently identifies architectures that outperform the baseline in all evaluation metrics. Compared to other zero-shot NAS methods, W-PCA leads in 6 out of 8 metrics for the OPT-1.3B size, all metrics for the OPT-2.7B size, and 7 out of 8 metrics for the LLaMA3.2-3B size. These results demonstrate the effectiveness of our method in transferring performance to CLM tasks.

