# OpenReview forum: "W-PCA Based Gradient-Free Proxy for Efficient Search of Lightweight Language Models"
_ICLR.cc/2025/Conference — ICLR 2025 Poster_

### Official Review · Reviewer_uDq6 · 2024-10-29

**Soundness:** 3
**Presentation:** 2
**Contribution:** 3
**Rating:** 6
**Confidence:** 2

**Summary:**

This paper introduce an approach that leverages a combination of principal component analysis (PCA) values and parameter counts as a zero-shot proxy metric to evaluate model candidates. By bypassing training and relying solely on initial weight information, W-PCA aims to efficiently estimate the potential of each candidate architecture, with significant gains in computational efficiency.

**Strengths:**

1. The paper provides an empirical evidence in Figure 2, showing that initial PCA scores of BERT-based candidate models correlate with their final performance, which is an interesting and effective shortcut for evaluating model quality.
2. The paper compares W-PCA with other zero-shot and one-shot NAS methods across benchmarks such as GLUE and SQuAD, demonstrating competitive performance. Moreover, the latency is drastically reduced.

**Weaknesses:**

1. While the paper presents a correlation between PCA values of initial weights and final performance, this correlation may be specific to the BERT architecture and the chosen search space. It remains unclear if this method would generalize well to other model architectures, such as GPT models or non-BERT Transformer variants.
2. The paper show curves of PCA scores over different training epoch in Figure 3 as a motivation of using PCA in the NAS, but it is better to also show a curve of accuracy besides epoch to better claim "Tracking performance via PCA".
3. The intuition behind combining PCA scores with parameter count to form the W-PCA metric is not well explained. While Figure 2 suggests that W-PCA achieves some gains, the performance patterns are not significantly more distinct than those observed with vanilla PCA.
4. In the ablation study, W-PCA shows only marginal improvements over baseline methods. Providing statistical significance tests on these improvements would be beneficial.

**Questions:**

See weaknesses.

---

> ### Author Response · Authors · 2024-11-21
> **Rebuttal**
>
> **W1.**
>
> We have conducted experiments on generative models from the OPT and LLaMA series, and the results are as follows:
>
> | Teacher      | Proxy                 | #Params | DollyEval |          | SelfInst |          | VicunaEval |          | S-NI     | UnNI     |
> | ------------ | --------------------- | ------- | --------- | -------- | -------- | -------- | ---------- | -------- | -------- | -------- |
> |              |                       |         | GPT4      | R-L      | GPT4     | R-L      | GPT4       | R-L      | R-L      | R-L      |
> | OPT-13B      | OPT-1.3B(baseline)    | -       | 60.7      | 26.7     | 47.0     | 14.8     | 50.6       | 17.9     | 28.6     | 33.4     |
> |              | Synaptic Diversity    | 1.3B    | 60.4      | 26.9     | 47.3     | 15.2     | 51.8       | 17.6     | 29.1     | 34.2     |
> |              | Head Confidence       | 1.3B    | 61.5      | 27.3     | **48.0** | 15.3     | 51.5       | 18.1     | **29.5** | 34.3     |
> |              | Softmax Confidence    | 1.3B    | 61.2      | 27.1     | 47.5     | 14.6     | 50.8       | 18.2     | 28.9     | 33.9     |
> |              | W-PCA                 | 1.3B    | **62.0**  | **27.6** | 47.9     | **16.1** | **52.3**   | **19.0** | 29.2     | **35.0** |
> | OPT-13B      | OPT-2.7B(baseline)    | -       | 63.2      | 27.4     | 52.7     | 17.2     | 55.9       | 19.1     | 30.7     | 35.1     |
> |              | Synaptic Diversity    | 2.7B    | 63.8      | 27.5     | 52.9     | 17.8     | 56.2       | 18.9     | 31.5     | 35.8     |
> |              | Head Confidence       | 2.7B    | 64.0      | 27.8     | 53.2     | 17.5     | 56.5       | 19.4     | 31.6     | 35.9     |
> |              | Softmax Confidence    | 2.7B    | 63.2      | 27.8     | 53.0     | 17.3     | 56.1       | 19.2     | 31.0     | 35.4     |
> |              | W-PCA                 | 2.7B    | **64.5**  | **28.3** | **53.9** | **18.1** | **57.2**   | **20.1** | **32.1** | **36.2** |
> | LLaMA3.2-11B | LLaMA3.2-3B(baseline) | -       | 74.0      | 29.8     | 69.1     | 23.6     | 64.6       | 25.2     | 36.2     | 43.9     |
> |              | Synaptic Diversity    | 3B      | 73.8      | 30.5     | 70.1     | 23.8     | 65.0       | 25.4     | 36.7     | 44.8     |
> |              | Head Confidence       | 3B      | 74.4      | 30.9     | 71.4     | 24.2     | 65.2       | 26.2     | 37.1     | **45.6** |
> |              | Softmax Confidence    | 3B      | 74.1      | 30.1     | 70.4     | 23.6     | 64.9       | 25.9     | 37.2     | 44.3     |
> |              | W-PCA                 | 3B      | **75.1**  | **31.4** | **71.8** | **24.7** | **65.9**   | **26.8** | **37.6** | 45.2     |
>
> It can be seen that W-PCA still demonstrates significant advantages in the search for generative models. We have included the experimental results and analysis in Appendix I.
>
> **W2.**
>
> We have added the curves showing the progression of the model's GLUE scores during training in Figure 3 and supplemented the caption and references with relevant explanations.
>
> **W3.**
>
> We have already cited [1] at the beginning of Section 3.3, noting that it is extremely challenging to discover or design a proxy that outperforms #Params in terms of stability and performance. After identifying PCA as an excellent proxy, multiplying it with #Params is a worthwhile attempt (as the additional computation time for a proxy is negligible compared to training neural networks). We have revised the opening sentences of Section 3.3 to clarify this explanation.
>
> Figure 2: Yes, compared to the improvements W-PCA brings over vanilla PCA, the improvements vanilla PCA offers over other methods are more significant. However, vanilla PCA itself is also one of our proposed innovations.
>
> **W4.**
>
> We conducted the relevant experiments, and the results are as follows:
>
> | Proxy   | GLUE      |
> | ------- | --------- |
> | #Params | 80.3±0.0  |
> | V-PCA   | 80.9±0.09 |
> | W-PCA   | 81.5±0.14 |
>
> We have added these results and the experimental setup to Section H.1. Please refer to our revised version for details.
>
>
>
> [1] Li G, Yang Y, Bhardwaj K, et al. ZiCo: Zero-shot NAS via inverse Coefficient of Variation on Gradients[C]//The Eleventh International Conference on Learning Representations.

---

> > ### Comment · Reviewer_uDq6 · 2024-11-21
> >
> > Thanks for the response and providing additional experiment results. I have raised my score.

---

> > > ### Author Response · Authors · 2024-11-22
> > >
> > > Thank you for supporting.

---

### Official Review · Reviewer_8SHz · 2024-11-04

**Soundness:** 3
**Presentation:** 3
**Contribution:** 3
**Rating:** 8
**Confidence:** 2

**Summary:**

The paper introduces a new zero-shot approach for NAS of lightweight language models, using a score based on PCA dimensionality and the model's parameter count. Experimental results demonstrate that this method outperforms prior NAS approaches and manually designed lightweight architectures.

**Strengths:**

The presentation is clear and well-written. The method outperforms previous baselines, is computationally efficient, and is grounded in solid foundational insights and preliminary observations.

**Weaknesses:**

**Major Weaknesses:**
1. The models used are somewhat outdated, as BERT is rarely a focus in current research, which has largely shifted toward GPT-style models.
2. Reproducibility is challenging without the provision of code, making it difficult for others to replicate the study’s findings.
3. The procedure is a bit unclear. Are the architectures tuned during genetic search
4. Are the other baselines fine-tuned? Do they employ Knowledge Distillation? If not, the comparison may be unfair.


**Minor Weaknesses:**

1. The term "PCA Value" is misleading, as PCA is primarily a dimensionality reduction method. It should be introduced with a brief explanation in the introduction and abstract, before discussing the preliminary experiments.
2. Table 5 is unclear—does it suggest that using only the parameter count as a proxy achieves better results than all previous methods?

**Questions:**

I suggest moving the Zero-Shot NAS section to the experimental settings to provide a clearer explanation of the baselines.

Additionally, Figure 2 at the beginning needs much more explanation, as it's difficult to understand the main idea without reading the entire paper. The figure illustrating the method could also be improved, as it currently doesn’t contribute to comprehension effectively.




**Overall**, I find the paper to be a good quality and ready to reconsider the score, once the authors address all the raised concerns.

---

> ### Author Response · Authors · 2024-11-21
> **Rebuttal**
>
> **W1.**
>
> Thank you, we have conducted experiments on generative models from the OPT and LLaMA series, and the results are as follows:
>
> | Teacher      | Proxy                 | #Params | DollyEval |          | SelfInst |          | VicunaEval |          | S-NI     | UnNI     |
> | ------------ | --------------------- | ------- | --------- | -------- | -------- | -------- | ---------- | -------- | -------- | -------- |
> |              |                       |         | GPT4      | R-L      | GPT4     | R-L      | GPT4       | R-L      | R-L      | R-L      |
> | OPT-13B      | OPT-1.3B(baseline)    | -       | 60.7      | 26.7     | 47.0     | 14.8     | 50.6       | 17.9     | 28.6     | 33.4     |
> |              | Synaptic Diversity    | 1.3B    | 60.4      | 26.9     | 47.3     | 15.2     | 51.8       | 17.6     | 29.1     | 34.2     |
> |              | Head Confidence       | 1.3B    | 61.5      | 27.3     | **48.0** | 15.3     | 51.5       | 18.1     | **29.5** | 34.3     |
> |              | Softmax Confidence    | 1.3B    | 61.2      | 27.1     | 47.5     | 14.6     | 50.8       | 18.2     | 28.9     | 33.9     |
> |              | W-PCA                 | 1.3B    | **62.0**  | **27.6** | 47.9     | **16.1** | **52.3**   | **19.0** | 29.2     | **35.0** |
> | OPT-13B      | OPT-2.7B(baseline)    | -       | 63.2      | 27.4     | 52.7     | 17.2     | 55.9       | 19.1     | 30.7     | 35.1     |
> |              | Synaptic Diversity    | 2.7B    | 63.8      | 27.5     | 52.9     | 17.8     | 56.2       | 18.9     | 31.5     | 35.8     |
> |              | Head Confidence       | 2.7B    | 64.0      | 27.8     | 53.2     | 17.5     | 56.5       | 19.4     | 31.6     | 35.9     |
> |              | Softmax Confidence    | 2.7B    | 63.2      | 27.8     | 53.0     | 17.3     | 56.1       | 19.2     | 31.0     | 35.4     |
> |              | W-PCA                 | 2.7B    | **64.5**  | **28.3** | **53.9** | **18.1** | **57.2**   | **20.1** | **32.1** | **36.2** |
> | LLaMA3.2-11B | LLaMA3.2-3B(baseline) | -       | 74.0      | 29.8     | 69.1     | 23.6     | 64.6       | 25.2     | 36.2     | 43.9     |
> |              | Synaptic Diversity    | 3B      | 73.8      | 30.5     | 70.1     | 23.8     | 65.0       | 25.4     | 36.7     | 44.8     |
> |              | Head Confidence       | 3B      | 74.4      | 30.9     | 71.4     | 24.2     | 65.2       | 26.2     | 37.1     | **45.6** |
> |              | Softmax Confidence    | 3B      | 74.1      | 30.1     | 70.4     | 23.6     | 64.9       | 25.9     | 37.2     | 44.3     |
> |              | W-PCA                 | 3B      | **75.1**  | **31.4** | **71.8** | **24.7** | **65.9**   | **26.8** | **37.6** | 45.2     |
>
> It can be seen that W-PCA still demonstrates significant advantages in the search for generative models. We have included the experimental results and analysis in Appendix I.
>
> **W2.**
>
> We have uploaded our implementation of W-PCA.
>
> **W3.**
>
> As shown in Figure 4, the role of the genetic algorithm is to encode the combinations of architectures within the search space and, after several generations of selection, crossover, and mutation (Appendix E), identify the optimal combination (i.e., the one with the highest W-PCA value). Once this optimal architecture is determined, knowledge distillation is used to train the weights, while the architecture itself remains unchanged.
>
> **W4.**
>
> As stated in Section 1 of Related Work, all manually designed neural networks in the baselines are trained using KD. NAS-BERT and EfficientBERT, compared to our method, adopt KD along with task-specific fine-tuning for training. AutoBERT-Zero trains models from scratch but employs a from-scratch training approach during the architecture search phase. Therefore, we only compare the NAS phase rather than the total training time.
>
> **w1:**
>
> We have rewritten the relevant statements in abstract and introduction.
>
> **w2:**
>
> This is expected. As stated in [1], designing a proxy with higher stability than #Params is quite challenging. Similarly, the study in [2] does not compare against #Params either.
>
> **Qs:**
>
> We provided further explanation in the caption of Figure 2 and in the text where Figure 2 is referenced, as well as in the caption of Figure 4.
>
>
>
> [1] Li G, Yang Y, Bhardwaj K, et al. ZiCo: Zero-shot NAS via inverse Coefficient of Variation on Gradients[C]//The Eleventh International Conference on Learning Representations.
>
> [2] Serianni A, Kalita J. Training-free Neural Architecture Search for RNNs and Transformers[C]//Proceedings of the 61st Annual Meeting of the Association for Computational Linguistics. 2023: 2522-2540.

---

> > ### Comment · Reviewer_8SHz · 2024-11-22
> > **Reply to rebuttal**
> >
> > Thank you for your extensive experiments and provided discussion. My concerns are fully addressed, I have raised the score.

---

> ### Author Response · Authors · 2024-11-23
> **Reply**
>
> Thank you for raising the score. It is satisfactory for us authors to see the work being recognized.

---

### Official Review · Reviewer_PoWa · 2024-11-04

**Soundness:** 3
**Presentation:** 3
**Contribution:** 4
**Rating:** 8
**Confidence:** 4

**Summary:**

The authors present a gradient-free method of evaluation of performance of candidate neural networks as part of neural architecture search. Their approach relies on estimating model's performance on target dataset through the PCA-score of FFN layers weighted by the number of parameters. The benefits of this approach include: strong correlation with target metric, gradient-free approach and the need for only a single forward pass on candidate network. Authors conduct experiment to evalaute the proposed method by sampling and evaluating 500 structures from FlexiBERT benchmark. The structures are evaluated as part of student model in distillation setup. The datasets are GLUE and SQuAD.

**Strengths:**

1) Authors propose a simple yet effective-enough method for evaluating candidate networks during NAS. It helps achieve networks with lowest latency and high performance on target datasets.
2) The method itself is highly efficient and allows for reasonably fast NAS iterations without compromising on quality as shown in Table 1.

**Weaknesses:**

1) Parts of paper appear to be LLM-written (for instance: L281-L304). Itself it is not a bad thing, yet those parts of paper lack substance and sometimes repeat the same information.
2) This paper would benefit from more experiments on decoder models (in addition to existing experiments on encoder models). This would make the paper even more relevant. Indeed, in the introduction, authors cite OpenAI's techincal report as evidence that LLMs have shown substantial performance across domains. It would be naturally to include said LLMs (maybe not OpenAI's LLMs, but generally) into the scope of this paper.
3) A reasonably-sized limitations section would also benefit the paper.

**Questions:**

1) L075-L081 - a lot of empty space here
2) Figure 3 - legend is very small, not easy to read
3) Do you always compute PCA scores at epoch 0, i.e right after models parameters are randomly initialized?
4) L461 - Actual values of reduced CO2 footprint would be more convincing.
5) Table 1 and subsection 5.2 - I suggest clarifying between what values do you compute correlation.
6) If answer to 3 is yes: does W-PCA score depends on the random seed that is used during initialization? If so, have you considered computing W-PCA as an average of scores from different initializations?

---

> ### Author Response · Authors · 2024-11-21
> **Rebuttal**
>
> **W1.**
>
> We have removed the redundant parts.
>
> **W2.**
>
> We conducted experiments on generative models from the OPT and LLaMA series, and the results are as follows:
>
> | Teacher      | Proxy                 | #Params | DollyEval |          | SelfInst |          | VicunaEval |          | S-NI     | UnNI     |
> | ------------ | --------------------- | ------- | --------- | -------- | -------- | -------- | ---------- | -------- | -------- | -------- |
> |              |                       |         | GPT4      | R-L      | GPT4     | R-L      | GPT4       | R-L      | R-L      | R-L      |
> | OPT-13B      | OPT-1.3B(baseline)    | -       | 60.7      | 26.7     | 47.0     | 14.8     | 50.6       | 17.9     | 28.6     | 33.4     |
> |              | Synaptic Diversity    | 1.3B    | 60.4      | 26.9     | 47.3     | 15.2     | 51.8       | 17.6     | 29.1     | 34.2     |
> |              | Head Confidence       | 1.3B    | 61.5      | 27.3     | **48.0** | 15.3     | 51.5       | 18.1     | **29.5** | 34.3     |
> |              | Softmax Confidence    | 1.3B    | 61.2      | 27.1     | 47.5     | 14.6     | 50.8       | 18.2     | 28.9     | 33.9     |
> |              | W-PCA                 | 1.3B    | **62.0**  | **27.6** | 47.9     | **16.1** | **52.3**   | **19.0** | 29.2     | **35.0** |
> | OPT-13B      | OPT-2.7B(baseline)    | -       | 63.2      | 27.4     | 52.7     | 17.2     | 55.9       | 19.1     | 30.7     | 35.1     |
> |              | Synaptic Diversity    | 2.7B    | 63.8      | 27.5     | 52.9     | 17.8     | 56.2       | 18.9     | 31.5     | 35.8     |
> |              | Head Confidence       | 2.7B    | 64.0      | 27.8     | 53.2     | 17.5     | 56.5       | 19.4     | 31.6     | 35.9     |
> |              | Softmax Confidence    | 2.7B    | 63.2      | 27.8     | 53.0     | 17.3     | 56.1       | 19.2     | 31.0     | 35.4     |
> |              | W-PCA                 | 2.7B    | **64.5**  | **28.3** | **53.9** | **18.1** | **57.2**   | **20.1** | **32.1** | **36.2** |
> | LLaMA3.2-11B | LLaMA3.2-3B(baseline) | -       | 74.0      | 29.8     | 69.1     | 23.6     | 64.6       | 25.2     | 36.2     | 43.9     |
> |              | Synaptic Diversity    | 3B      | 73.8      | 30.5     | 70.1     | 23.8     | 65.0       | 25.4     | 36.7     | 44.8     |
> |              | Head Confidence       | 3B      | 74.4      | 30.9     | 71.4     | 24.2     | 65.2       | 26.2     | 37.1     | **45.6** |
> |              | Softmax Confidence    | 3B      | 74.1      | 30.1     | 70.4     | 23.6     | 64.9       | 25.9     | 37.2     | 44.3     |
> |              | W-PCA                 | 3B      | **75.1**  | **31.4** | **71.8** | **24.7** | **65.9**   | **26.8** | **37.6** | 45.2     |
>
> It can be seen that W-PCA still demonstrates significant advantages in the search for generative models. We have included the experimental results and analysis in Appendix I.
>
> **W3.**
>
> We have added this section before the references.
>
> **Q1.**
>
> We have adjusted these formats.
>
> **Q2.**
>
> We have enlarged the legend in Figure 3. Additionally, due to the insertion of a new figure, Figure 3 has been adjusted to display on a full page.
>
> **Q3 & Q6.**
>
> Yes. We have conducted the relevant experiments, and the results are as follows:
>
> | Proxy   | GLUE      |
> | ------- | --------- |
> | #Params | 80.3±0.0  |
> | V-PCA   | 80.9±0.09 |
> | W-PCA   | 81.5±0.14 |
>
> We have included these results and the experimental setup in Section H.1. Please refer to our revised version for details.
>
> **Q4.**
>
> We have included the reduction in $\ce{CO2}$ emissions in the footnote.
>
> **Q5.**
>
> We have added additional explanations to the caption of Table 1. Specifically, we compute the correlation between each zero-shot proxy and the ground truth performance of the neural networks, represented by their BLEU scores.

---

### Official Review · Reviewer_Z4ps · 2024-11-08

**Soundness:** 3
**Presentation:** 3
**Contribution:** 2
**Rating:** 5
**Confidence:** 3

**Summary:**

This paper introduces W-PCA, a novel zero-shot NAS method specifically tailored for light weight language models.

**Strengths:**

1.The method is simple but achieves both better latency and accuracy on NLU tasks.

**Weaknesses:**

1.The format of this paper is confusing. The gap between figures and text(e.g. Figure 2) seems to be abnormal. Table 4 is also not placed correctly.

2.Poor writing and low-quality figures. For example, I have no idea on the pipeline and implementation of W-PCA until the experiement section. the lines representing dimensions are not straight lines in figure 4. The comparison in the conclusion part(e.g. Kendall score and Spearman score) seems to be inconsistent with experiment results.

3.Although authors claim that W-PCA achieves SOTA, their results are based on their own version of BERT model. They compare their methods with others, whose results are directly copied from their original papers. The comparison is unfair and may mitigate the reliability of their results, especially on the GLUE benchmark.

4.The experiments only focus on BERT models and they only have are around 100M. This limits the applicability of W-PCA to current LLMs, which typically have several billion parameters.

**Questions:**

1.Could you provide results of some other methods based on your BERT models?

2.I am curious about the applicability of W-PCA on current LLMs(e.g. LLama).

3.How do you calculate GPU days in table 3 and estimate that AutoBERT-Zero-small requires around 1,000 GPU days on GLUE dev set?

4.You claim in the conclusion that W-PCA achieves a Kendall score and a Spearman Score that surpass previous methods by 0.207 and 0.325 respectively. However in table 1, the differences are 0.220 and 0.334. Could you clarify how do you calculate the score differences?

---

> ### Author Response · Authors · 2024-11-21
> **Rebuttal 1/2**
>
> **Q1 & W3.**
>
> Existing transformer-based zero-shot NAS methods (also known as training-free NAS methods, as summarized in L88 at original version) typically perform ranking correlation experiments on NAS benchmark datasets, without accuracy experiments on NLU datasets. As stated in L423-L427 and L457 (original version), for the sake of comparison, we placed the previous zero-shot NAS methods within the search space we designed. The results of the five zero-shot NAS methods in Table 2 and the three methods in Table 3 were all obtained within our custom search space.
>
> It is impractical to place training-based NAS methods (including Vanilla NAS, which retrains each neural network, and one-shot NAS, which samples from a supernet) and zero-shot NAS proxies in the same search space because the design of the search stage, search targets, search space, and training methods are integral to each training-based NAS approach. For example, EfficientBERT employs a coarse-to-fine three-stage search strategy, and both EfficientBERT and AutoBERT-Zero even include activation function searches. In training-free NAS, during the network search stage, the neural network can only use initialized weights, with training strictly prohibited. Performance is typically evaluated using a single forward pass with one batch of data (with an additional backward pass for non-gradient-free methods), making it impossible to compare training methods with training-based NAS.
>
> For training-based NAS methods, sampling within the search space is often just one phase. Changing the search space in a particular stage would render the comparison meaningless as it would alter the fundamental essence of these methods. Generally, the evaluation of different NAS methods is determined by comparing search time and the performance of the final models obtained. However, comparing different zero-shot NAS methods within the same search space is feasible. Therefore, in Section H.3 (revised version H.4), we also conducted a comparison of our approach with other zero-shot NAS methods within the EfficientBERT search space.
>
> **Q2 & W4.**
>
> We conducted experiments on generative models from the OPT and LLaMA series, and the results are as follows:
>
> | Teacher      | Proxy                 | #Params | DollyEval |          | SelfInst |          | VicunaEval |          | S-NI     | UnNI     |
> | ------------ | --------------------- | ------- | --------- | -------- | -------- | -------- | ---------- | -------- | -------- | -------- |
> |              |                       |         | GPT4      | R-L      | GPT4     | R-L      | GPT4       | R-L      | R-L      | R-L      |
> | OPT-13B      | OPT-1.3B(baseline)    | -       | 60.7      | 26.7     | 47.0     | 14.8     | 50.6       | 17.9     | 28.6     | 33.4     |
> |              | Synaptic Diversity    | 1.3B    | 60.4      | 26.9     | 47.3     | 15.2     | 51.8       | 17.6     | 29.1     | 34.2     |
> |              | Head Confidence       | 1.3B    | 61.5      | 27.3     | **48.0** | 15.3     | 51.5       | 18.1     | **29.5** | 34.3     |
> |              | Softmax Confidence    | 1.3B    | 61.2      | 27.1     | 47.5     | 14.6     | 50.8       | 18.2     | 28.9     | 33.9     |
> |              | W-PCA                 | 1.3B    | **62.0**  | **27.6** | 47.9     | **16.1** | **52.3**   | **19.0** | 29.2     | **35.0** |
> | OPT-13B      | OPT-2.7B(baseline)    | -       | 63.2      | 27.4     | 52.7     | 17.2     | 55.9       | 19.1     | 30.7     | 35.1     |
> |              | Synaptic Diversity    | 2.7B    | 63.8      | 27.5     | 52.9     | 17.8     | 56.2       | 18.9     | 31.5     | 35.8     |
> |              | Head Confidence       | 2.7B    | 64.0      | 27.8     | 53.2     | 17.5     | 56.5       | 19.4     | 31.6     | 35.9     |
> |              | Softmax Confidence    | 2.7B    | 63.2      | 27.8     | 53.0     | 17.3     | 56.1       | 19.2     | 31.0     | 35.4     |
> |              | W-PCA                 | 2.7B    | **64.5**  | **28.3** | **53.9** | **18.1** | **57.2**   | **20.1** | **32.1** | **36.2** |
> | LLaMA3.2-11B | LLaMA3.2-3B(baseline) | -       | 74.0      | 29.8     | 69.1     | 23.6     | 64.6       | 25.2     | 36.2     | 43.9     |
> |              | Synaptic Diversity    | 3B      | 73.8      | 30.5     | 70.1     | 23.8     | 65.0       | 25.4     | 36.7     | 44.8     |
> |              | Head Confidence       | 3B      | 74.4      | 30.9     | 71.4     | 24.2     | 65.2       | 26.2     | 37.1     | **45.6** |
> |              | Softmax Confidence    | 3B      | 74.1      | 30.1     | 70.4     | 23.6     | 64.9       | 25.9     | 37.2     | 44.3     |
> |              | W-PCA                 | 3B      | **75.1**  | **31.4** | **71.8** | **24.7** | **65.9**   | **26.8** | **37.6** | 45.2     |
>
> It can be seen that W-PCA still demonstrates significant advantages in the search for generative models. We have included the experimental results and analysis in Appendix I.

---

> ### Author Response · Authors · 2024-11-21
> **Rebuttal 2/2**
>
> **Q3.**
>
> As stated by literature [1] on page 10667, “ The searching phase costs around 24K GPU hours (760+ candidates) on Nvidia V100.”
>
> **Q4 & W2:.**
>
> We have fixed the mistake. We previously set the $\eta$ value to 0.99 for both the ranking evaluation and accuracy comparison experiments. In Appendix D, we discussed the robustness of different $\eta$ values and found that setting $\eta$ to 0.9 yields better results in the ranking evaluation experiment. As a result, we updated the experimental results in the tables and figures but forgot to update the text in the conclusion section. As can be seen in Table 9, the increases of 0.207 and 0.325 are data for which $\eta$ is 0.99.
>
> **W1.**
>
> We have adjusted the formatting so that Figure 2 and Table 4 are now displayed more normally.
>
> ​
>
> ***About writng:***
>
> We have revised the content of abstract and introduction, and expanded the caption of Figure 2 and Figure 4 to make the article more coherent. (For zero-shot NAS, the important thing is how to find the optimal network structure, and the training of the network structure is generally not in the method section.)
>
> In addition, the style of Figure 4 is borrowed from Figure 2 of the literature [1], where we have now straightened the lines.
>
>
>
> [1] Gao J, Xu H, Shi H, et al. Autobert-zero: Evolving bert backbone from scratch[C]//Proceedings of the AAAI Conference on Artificial Intelligence. 2022, 36(10): 10663-10671.

---

> > ### Comment · Reviewer_Z4ps · 2024-11-24
> >
> > Thank you for responding all my questions and providing additional experiment results. I have raised my score accordingly.

---

### Author Response · Authors · 2024-11-21

Dear reviewers,

​		Thank you for your valuable feedback! We have submitted the rebuttal.

---

### Meta-Review · Area_Chair_DvfV · 2024-12-19

**Metareview:**

The authors propose a gradient-free method for evaluating the performance of candidate neural networks within neural architecture search. This method estimates a model's performance on a target dataset by using the PCA-score of FFN layers, weighted by the number of parameters. Key advantages of this approach include a strong correlation with target metrics, a gradient-free design, and the ability to evaluate a candidate network with only a single forward pass. The authors validate their method through experiments involving 500 sampled structures from the FlexiBERT benchmark, tested as student models in a distillation setup on the GLUE and SQuAD datasets.

Strengths:

- W-PCA offers a straightforward yet powerful method that achieves competitive accuracy while reducing latency.
- The zero-shot nature of W-PCA drastically reduces computational requirements, making it practical for large-scale NAS tasks.

Weaknesses:
- The writing can be improved and the computational cost for search seems to be quite intensive.
- While W-PCA shows gains over vanilla PCA, these improvements are relatively modest in the ablation study.

**Additional Comments On Reviewer Discussion:**

The paper received a final score of 5886. The primary concern raised by the reviewer who gave a score of 5 was that the method was only tested on BERT. In the rebuttal, the authors provided results on three additional models, demonstrating that their proposed approach performs well. I found the additional experiments satisfactory.

---

### Decision · Program_Chairs · 2025-01-22

Accept (Poster)